# Antagonism of interferon signaling by fibroblast growth factors promotes viral replication

Luigi Maddaluno[1],[†],[*] (iD), Corinne Urwyler[1],[†], Theresa Rauschendorfer[1],[†], Michael Meyer[1], Debora Stefanova[1], Roman Spörri[2], Mateusz Wietecha[1], Luca Ferrarese[1], Diana Stoycheva[2], Daniela Bender[3], Nick Li[1],[4], Gerhard Strittmatter[4], Khondokar Nasirujjaman[1],[5] (iD), Hans-Dietmar Beer[4] (iD), Peter Staeheli[6],[7], Eberhard Hildt[3], Annette Oxenius[2] & Sabine Werner[1],[**] (iD)

## Abstract

Fibroblast growth factors (FGFs) play key roles in the pathogenesis of different human diseases, but the cross-talk between FGFs and other cytokines remains largely unexplored. We identified an unexpected antagonistic effect of FGFs on the interferon (IFN) signaling pathway. Genetic or pharmacological inhibition of FGF receptor signaling in keratinocytes promoted the expression of interferon-stimulated genes (ISG) and proteins *in vitro* and *in vivo*. Conversely, FGF7 or FGF10 treatment of keratinocytes suppressed ISG expression under homeostatic conditions and in response to IFN or poly(I:C) treatment. FGF-mediated ISG suppression was independent of IFN receptors, occurred at the transcriptional level, and required FGF receptor kinase and proteasomal activity. It is not restricted to keratinocytes and functionally relevant, since FGFs promoted the replication of herpes simplex virus I (HSV-1), lymphocytic choriomeningitis virus, and Zika virus. Most importantly, inhibition of FGFR signaling blocked HSV-1 replication in cultured human keratinocytes and in mice. These results suggest the use of FGFR kinase inhibitors for the treatment of viral infections.

**Keywords** fibroblast growth factor; FGF receptor; Herpes simplex virus; interferon; Zika virus

**Subject Categories** Pharmacology & Drug Discovery; Immunology; Microbiology, Virology & Host Pathogen Interaction

## Introduction

Fibroblast growth factors (FGFs), which comprise a family of 22 proteins, are master regulators of development and tissue repair, and abnormalities in FGF signaling are involved in the pathogenesis of major human diseases, including developmental and inflammatory diseases and cancer (Beenken & Mohammadi, 2009; Ornitz & Itoh, 2015). Most FGFs signal through one or more of four receptor tyrosine kinases, designated FGF receptor (FGFR)1 to FGFR4. Additional complexity is achieved by alternative splicing, resulting in the production of FGFR splice variants with different ligand binding specificities (Ornitz & Itoh, 2015). Expression of most FGFs is low under homeostatic conditions, but induced upon injury to various tissues and organs, which is essential for efficient tissue repair and regeneration (Maddaluno *et al*, 2017). Therefore, recombinant FGFs are clinically used for the promotion of tissue repair, such as cutaneous wound healing, and for the protection of tissues from different types of damage (Nunes *et al*, 2016; Zhang & Li, 2016). On the other hand, enhanced expression of FGFs or FGFRs and/or abnormal FGFR signaling is frequently observed in cancer, and FGFR inhibitors are therefore in clinical trials for the treatment of various malignancies (Tanner & Grose, 2016). Given these important activities of FGFs, it is paramount to identify the mechanisms of action of FGFs, the responsible target genes, and the cross-talk between FGFs and other growth factors and cytokines.

To study FGF function in the skin, we previously generated mice lacking FGFR1, FGFR2, or both receptors in keratinocytes of the epidermis and hair follicles. These mice develop a progressive

1 Department of Biology, Institute of Molecular Health Sciences, ETH Zurich, Zurich, Switzerland
2 Department of Biology, Institute of Microbiology, ETH Zurich, Zurich, Switzerland
3 Department of Virology, Paul-Ehrlich-Institute, Langen, Germany
4 Department of Dermatology, University Hospital of Zurich, Zurich, Switzerland
5 Department of Genetic Engineering and Biotechnology, University of Rajshahi, Rajshahi, Bangladesh
6 Institute of Virology, University Hospital Freiburg, Freiburg, Germany
7 Faculty of Medicine, University of Freiburg, Freiburg, Germany
*Corresponding author. Tel: +41 44 633 3941; E-mail: luigi.maddaluno@gmail.com
**Corresponding author. Tel: +41 44 633 3941; E-mail: Sabine.Werner@biol.ethz.ch
†These authors contributed equally to this work as first authors

inflammatory skin disease with features resembling atopic dermatitis in humans, which results at least in part from a defect in the epidermal barrier (Yang *et al*, 2010). However, the epidermal and immune phenotype only develops around 3 weeks after birth, and therefore, young mice are ideally suited for the identification of direct targets of FGF signaling in the epidermis.

Here, we show that expression of various ISGs is suppressed by FGFs in keratinocytes and correlates with enhanced replication of different types of viruses. These results identify FGF receptor inhibitors as promising compounds for antiviral therapies.

# Results

## Inhibition of FGFR signaling promotes expression of IFN-stimulated genes in keratinocytes

To identify FGF targets in the epidermis, we performed microarray analysis of epidermal RNA from mice lacking FGFR1 and FGFR2 in keratinocytes (K5-R1/R2 mice) and control mice without Cre recombinase (Yang *et al*, 2010) at postnatal day 18 (P18) and thus prior to the development of skin inflammation and epidermal thickening. Functional enrichment analysis of the differentially regulated genes ($\log_2$(f.c.) > 1.5, $P < 0.05$) identified enriched ontologies and pathways that reflect the epidermal abnormalities of K5-R1/R2 mice and the loss of hair follicles and associated melanocytes (Table 1A and B). Remarkably, the top hits are genes characteristic for the "Type I Interferon Response" (Table 1A and B, and Dataset EV1), and at least 15% (11/63) of the genes that were upregulated in K5-R1/R2 epidermis compared with controls ($\log_2$(f.c.) > 1.5, $P < 0.05$), in particular the majority of the most strongly regulated genes, are classical IFN-stimulated genes (ISGs) (Table 1C). Upregulation of ISGs was confirmed by qRT–PCR analysis of epidermal RNA from adult mice for some of the genes identified in the microarray and for additional ISGs, including IFN response factor 7 (*Irf7*), radical S-adenosyl methionine domain containing 2 (*Rsad2*), 2′-5′-oligoadenylate synthetase-like 2 (*Oasl2*), IFN-induced protein with tetratricopeptide repeat 1 (*Ifit1*), signal transducer and activator of transcription 1 (*Stat1*), and *Stat2* (Fig 1A). Upregulation of some ISGs was already significant at day 5 after birth (P5) and very robust at P9 (Fig 1B) and thus prior to the development of the barrier defect and associated cutaneous abnormalities and concomitant with the complete deletion of the *Fgfr1* and *Fgfr2* genes (Yang *et al*, 2010; Sulcova *et al*, 2015).

Elevated expression of IRF7 in the epidermis of adult K5-R1/R2 mice was confirmed by strong immunostaining in the nuclei of basal and suprabasal keratinocytes of the mutant mice (Fig 1C, red dots indicated by arrows). The nuclear localization of IRF7, which together with IRF3 and NF-κB controls the expression of interferons type I (IFN-α and IFN-β) and type III (IFN-λ), hints at enhanced production of IFNs in the epidermis of FGFR1/R2-deficient mice. Indeed, *Ifnb* and *Ifnl3* expression was significantly increased in freshly isolated, non-cultured keratinocytes from the epidermis of K5-R1/R2 mice, concomitantly with all tested ISGs. This was already seen at P7 and thus preceding the increase in immune cells (Fig 1D).

Upregulation of ISG expression is a cell-autonomous effect, since it was also observed in cultured primary keratinocytes from neonate K5-R1/R2 mice (Fig 1E). However, it was less pronounced

compared with non-cultured, freshly isolated keratinocytes (P7), suggesting that IFNs produced by immune cells *in vivo* potentiate this effect. An effect of Cre recombinase on ISG expression was excluded, since the expression levels of these genes were similar in primary keratinocytes isolated from the epidermis of wild-type and K5-Cre mice (Fig 1F). Upregulation of ISG expression in the absence of FGFR1 and FGFR2 was confirmed at the protein level for IRF1, IRF7, and IRF9 by Western blot analysis of total and in particular of nuclear lysates of spontaneously immortalized keratinocytes from K5-R1/R2 mice (Fig 1G).

The important role of FGFR signaling in the suppression of ISG expression was confirmed when human immortalized keratinocytes (HaCaT cells) were treated for 48 h with the FGFR kinase inhibitor BGJ398 (Guagnano *et al*, 2011) in the presence of serum, which contains low levels of FGFs. FGFR inhibition promoted expression of ISGs, including *RSAD2* and interferon-stimulated gene 15 (*ISG15*) (Fig 1H).

## FGFs suppress ISG expression in keratinocytes and intestinal epithelial cells

To further determine whether FGF signaling directly regulates ISG expression, we treated primary murine keratinocytes with FGF7, a major ligand of the FGFR2 splice variant expressed on keratinocytes (FGFR2b) (Ornitz & Itoh, 2015). Like many other cell types (Gough *et al*, 2012), mouse keratinocytes show a tonic expression of various ISGs. Treatment of these cells with FGF7 for only 3–6 h suppressed the tonic expression of all ISGs that we tested (Fig 2A). This was verified with HaCaT keratinocytes at the RNA level for ISGs, which were also regulated in mouse keratinocytes and which have a human orthologue with similar function (Fig 1 and Table 1C) and also for additional ISGs that we tested with the human cells. In HaCaT cells, FGF7 suppressed, for example, the expression of *IRF7*, *RSAD2,* and *IRF1* (Figs 2B and EV1A) and of suppressor of cytokine signaling 1 and 3 (*SOCS1* and *SOCS3*) (Fig EV1A), which are ISG products and involved in negative regulation of IFN signaling (Blumer *et al*, 2017; Michalska *et al*, 2018). FGF10, which also activates FGFR2b and in addition FGFR1b (Ornitz & Itoh, 2015), had a similar effect (Figs 2B and EV1A). The effect of FGF7 was verified at the protein level for IRF1 and IRF9 (Fig 2C, see also Fig 3C). By contrast, IRF3 expression was not suppressed by FGF7 at the RNA or protein level (Figs EV1A and 2C). The efficiency of FGF7 and FGF10 treatment in these experiments was validated by analysis of the expression of dual-specific phosphatase 6 (*DUSP6*), a classical FGF target gene that is strongly upregulated in response to FGFs (Li *et al,* 2007; Fig EV1A).

The suppressive effect of FGF7 on ISG expression and the concomitant increase in *DUSP6* expression was also observed with the Caco-2 colon cancer cell line, which is responsive to FGF7 (Visco *et al*, 2009). While *IRF7* expression was not regulated by FGF7 in these cells, there was a significant FGF7-induced reduction of the mRNA levels of IFIT1, ISG15, and also of the pattern recognition receptor RIG-I [retinoic acid inducible gene I (encoded by the *DDX58* gene)], which upon ligand binding promotes IFN expression (Bruns & Horvath, 2014; Fig EV1B). RSAD2 mRNA was not detectable in these cells.

The effect of FGF7 on ISG expression was dependent on the FGFR kinase activity, since it was largely inhibited in primary mouse keratinocytes or human HaCaT keratinocytes upon treatment with the FGFR kinase inhibitors AZD4547 (Gavine *et al*, 2012) or

**Table 1. Type I interferon signaling is the top regulated pathway in the epidermis of K5-R1/R2 mice.**

| (A) Gene Ontology (GO) Biological Process 2017b | | | |
|---|---|---|---|
| Index | Name | *P*-value | Adj *P*-value |
| 1 | Negative regulation of single stranded viral RNA replication via double-stranded DNA intermediate (GO:0045869) | 0.000008 | 0.001836 |
| 2 | Negative regulation of viral genome replication (GO:0045071) | 0.000006 | 0.001836 |
| 3 | Negative regulation by host of viral genome replication (GO:0044828) | 0.000015 | 0.002497 |
| 4 | Melanin biosynthetic process from tyrosine (GO:0006583) | 0.000000 | 0.000315 |
| 5 | Skin epidermis development (GO:0098773) | 0.000171 | 0.006386 |
| 6 | Defense response to fungus (GO:0050832) | 0.000379 | 0.006386 |
| 7 | Antifungal innate immune response (GO:0061760) | 0.000379 | 0.006386 |
| 8 | Defense response to fungus, incompatible interaction (GO:0009817) | 0.000379 | 0.006386 |
| 9 | Neutrophil-mediated killing of fungus (GO:0070947) | 0.000379 | 0.006386 |
| 10 | Negative regulation of single stranded viral RNA replication via double-stranded DNA intermediate (GO:0045869) | 0.000008 | 0.001836 |

| (B) Reactome 2016 | | | |
|---|---|---|---|
| Index | Name | *P*-value | Adj *P*-value |
| 1 | Interferon alpha/beta signaling_Homo sapiens_R-HSA-909733 | 0.000017 | 0.001495 |
| 2 | Interferon Signaling_Homo sapiens_R-HSA-913531 | 0.000870 | 0.039131 |
| 3 | Glutathione conjugation_Homo sapiens_R-HSA-156590 | 0.003484 | 0.104528 |
| 4 | Antiviral mechanism by IFN-stimulated genes_Homo sapiens_R-HSA-1169410 | 0.011532 | 0.207580 |
| 5 | ISG15 antiviral mechanism-Homo sapiens_R-HSA-1169408 | 0.011532 | 0.207580 |

| (C) ISGs that are upregulated ($\log_2$(f.c.) > 1.5, *P* < 0.05) in the epidermis of K5-R1/R2 mice at P18 (microarray data) | | | | | |
|---|---|---|---|---|---|
| Gene Symbol | Description | Gene ID | Affy Probe ID | $\log_2$(f.c.) | *P*-value |
| *Rtp4* | Receptor transporter protein 4 | 67775 | 1418580_at | 2.841 | 0.003 |
| *Rsad2* | Radical S-adenosyl methionine domain containing 2 | 58185 | 1421009_at | 2.824 | 0.007 |
| *Il1f8* | Interleukin 1 family, member 8 | 69677 | 1425715_at | 2.652 | 0.014 |
| *Ifit1* | Interferon-induced protein with tetratricopeptide repeats 1 | 15957 | 1450783_at | 2.649 | 0.014 |
| *Rsad2* | Radical S-adenosyl methionine domain containing 2 | 58185 | 1436058_at | 2.589 | 0.021 |
| *Ifi27l2a* | Interferon, alpha-inducible protein 27 like 2A | 76933 | 1426278_at | 2.506 | 0.021 |
| *Ifi44* | Interferon-induced protein 44 | 99899 | 1423555_a_at | 2.493 | 0.004 |
| *Isg15* | ISG15 ubiquitin-like modifier | 100038882 | 1431591_s_at | 2.027 | 0.009 |
| *Rsad2* | Radical S-adenosyl methionine domain containing 2 | 58185 | 1421008_at | 1.931 | 0.009 |
| *Oas1d* | 2′-5′-oligoadenylate synthetase 1D | 100535 | 1416847_s_at | 1.879 | 0.039 |
| *Defb3* | Defensin beta 3 | 27358 | 1421806_at | 1.56 | 0.044 |

BGJ398 (Guagnano *et al*, 2011; Fig EV2A and B). Blockade of the major FGFR signaling pathways, the phosphatidylinositide 3-kinase (PI3K)/AKT and MEK1/2-ERK1/2 pathways, mildly, but non-significantly, reduced the effect of FGF7 on the expression of some ISGs in HaCaT keratinocytes, while other ISGs were not affected (Fig EV2C). Importantly, however, combined treatment with MEK1/2 and PI3K inhibitors partially or completely blocked the effect of FGF7 (Fig EV2C). By contrast, we did not observe an effect of phospholipase Cγ inhibition (Fig EV2D). Verification of the activity of the inhibitors is shown in Appendix Fig S1.

In contrast to ISGs, IFN expression is hardly detectable in keratinocytes under normal culture conditions, and therefore, it is unlikely that a further reduction in IFN protein levels contributes to the effect of FGF7. Furthermore, expression of *SOCS1* and *SOCS3*,

which inhibit IFN signaling by competing with STATs for receptor binding and by suppression of JAK activity (Ivashkiv & Donlin, 2014), was even down-regulated by FGF7 or FGF10 (Fig EV1A). Rather, the rapid suppression of the expression level of several ISGs suggests that FGF7 regulates ISGs independent of IFN receptors. To test this possibility, we treated primary keratinocytes from mice lacking the IFN-α and IFN-β receptor (IFNAR) or double knockout mice deficient in IFNAR and the IFN-λ receptor (IFNLR) with FGF7. As expected, expression levels of ISGs were significantly lower in knockout compared to wild-type cells (Fig 2D), but still detectable (Cp values around 27.5 in the cells from double knockout mice). Nevertheless, the FGF7-induced suppression of ISG expression was still observed in the absence of these receptors (Fig 2E). These findings strongly suggest that FGF7 signaling

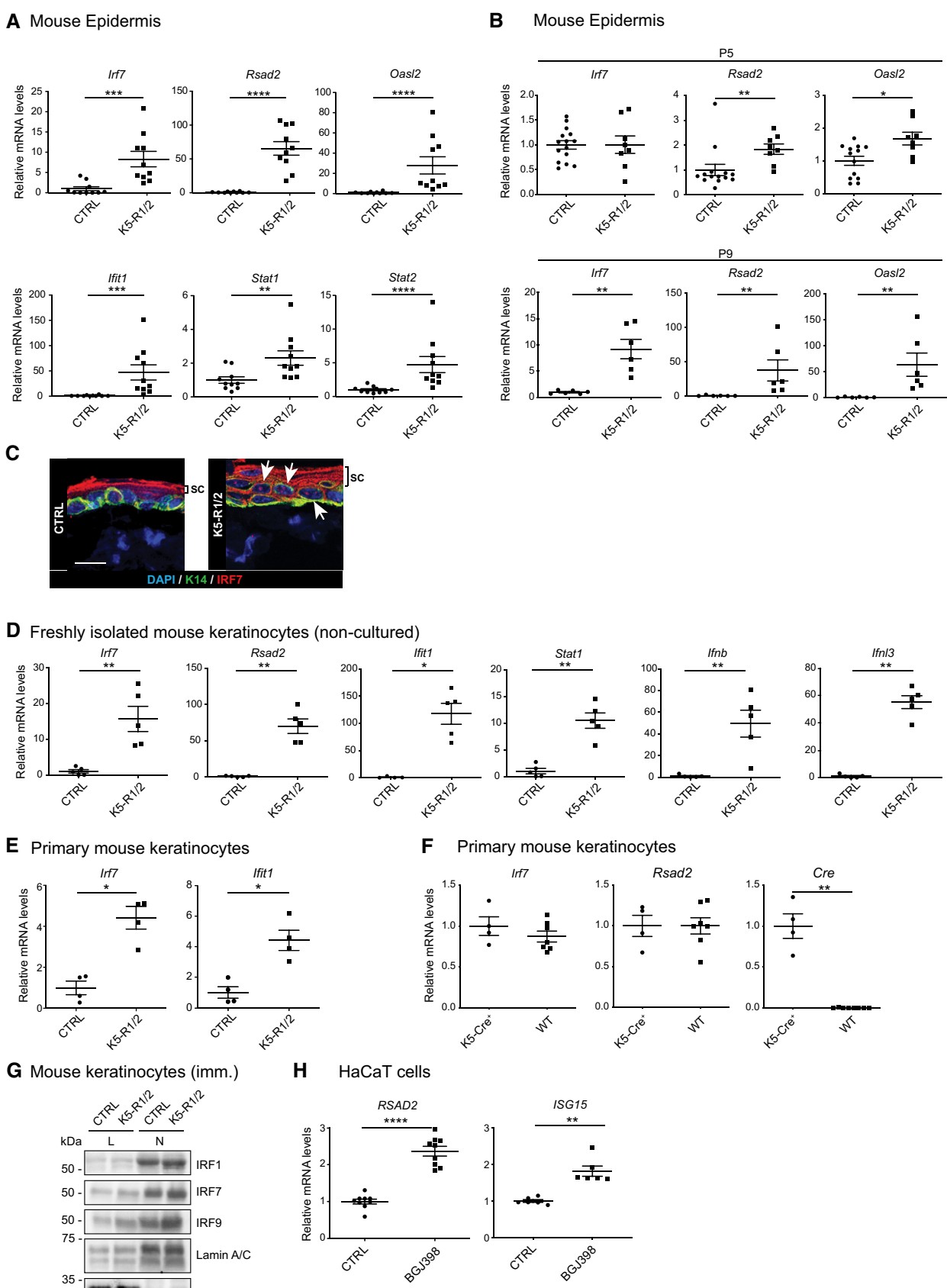

**Figure 1.**

**Figure 1. Inhibition of FGFR signaling in keratinocytes promotes IFN and ISG expression.**

A, B  qRT–PCR for different ISGs relative to *Rps29* using epidermal RNA from 8-week-old (A) or 5-day-old or 9-day-old (B) K5-R1/R2 and control (CTRL) mice.

C  Confocal microscopy images of epidermal sheets from back skin of K5-R1/R2 and CTRL mice stained for IRF7 and K14, counterstained with DAPI. Arrows denote nuclear IRF7 (red) in basal and suprabasal keratinocytes of K5-R1/R2 mice. The strong red staining of the *stratum corneum* (sc) is unspecific background and is more pronounced in K5-R1/R2 mice due to the increased thickness of this layer.

D  qRT–PCR for different ISGs, *Ifnb*, and *Ifnl3* relative to *Rps29* using RNA from freshly isolated, non-cultured keratinocytes of K5-R1/R2 and control (CTRL) mice at P7.

E  qRT–PCR for ISGs relative to *Rps29* using RNA from primary keratinocytes of neonatal CTRL and K5-R1/R2 mice.

F  qRT–PCR for ISGs and Cre relative to *Rps29* using RNA from primary keratinocytes of neonatal K5-Cre and wild-type (WT) mice.

G  Western blot analysis of total (L) and nuclear (N) lysates from immortalized (imm.) keratinocytes of CTRL and K5-R1/R2 mice for IRF1, IRF7, IRF9, Lamin A/C (nuclear marker, loading control), and GAPDH (cytosolic marker, loading control).

H  qRT–PCR for *RSAD2* and *ISG15* relative to *RPLP0* using RNA from HaCaT keratinocytes treated for 48 h with the FGFR inhibitor BGJ398 (3.5 μM) or vehicle in the presence of serum.

Data information: Scatter plots show mean ± SEM. Mean expression levels in CTRL mice or cell cultures were set to 1, and mRNA levels relative to this value are shown. In (F), expression levels in K5-Cre mice were set to 1. (A) $N = 10$ mice per genotype, (B) $N = 6$–14 mice per genotype. (C) Representative images of two experiments. Magnification bar: 20 μm. (D) $N = 5$ mice per genotype. (E, F) $N = 4$–7 cultures (each from a different mouse) per genotype. (G) Representative of three experiments. (H) $N = 6$–9 per treatment group. *$P \leq 0.05$, **$P \leq 0.01$, ***$P \leq 0.001$, ****$P \leq 0.0001$ (Mann–Whitney *U*-test). Exact *P*-values are provided in Dataset EV2. Source data are available online for this figure.

directly affects the transcription of ISGs. Consistent with this assumption, FGF7 suppressed the basal expression level of a luciferase reporter gene driven by five interferon-stimulated response elements (ISRE) in HaCaT cells (Fig 2F). Since the effect of FGF7 is rapid, it may be the consequence of FGF7-mediated degradation of a protein required for efficient ISG transcription. Indeed, the effect of FGF7 was abolished in the presence of the proteasome inhibitors MG132, epoxomicin, or bortezomib (Fig 2G and H, and Appendix Fig S2A). The activity of the proteasome inhibitors was verified by Western blot analysis of the nuclear factor erythroid 2-related factor 2 (NRF2) transcription factor, which is rapidly degraded by the proteasome under normal conditions (Villeneuve *et al*, 2010; Appendix Fig S2B).

**FGF7 suppresses IFN-induced ISG expression**

We next tested whether FGF7 also suppresses the response of keratinocytes to exogenous IFNs. Indeed, the IFN-α-induced upregulation of ISG expression in HaCaT cells was at least in part suppressed by

FGF7 (Fig 3A). FGF7 exerted this effect in a dose-dependent manner and was effective in the range between 1 and 10 ng/ml (Fig 3B).

Western blot analysis showed strong suppression of IRF1 protein levels by FGF7 (12 h treatment) in the presence or absence of IFN-α. However, levels of total and phosphorylated STAT1 (Y701 and S727) and of RSAD2 were not affected, possibly because of a longer half-life of these proteins. IRF9 protein levels were down-regulated by FGF7 in the absence, but not in the presence of IFN-α (Fig 3C).

The suppressive effect of FGF7 on the basal and IFN-α-induced ISG expression was verified with primary human keratinocytes (Fig 3D). We only used cells from two donors in this experiment, but the values were very similar among them.

**FGF7 suppresses poly(I:C)-induced ISG and IFN expression**

ISGs encode various proteins that control IFN production, such as the transcription factors IRF7 and IRF1 (Honda *et al*, 2005; Panda *et al*, 2019), and different proteins involved in nucleic acid sensing (Shaw *et al*, 2017). Therefore, we tested if FGF signaling also suppresses IFN expression. When HaCaT cells were treated with

**Figure 2. FGF7 and FGF10 suppress the IFN response in keratinocytes.**

A  Growth factor-starved primary keratinocytes from WT mice were treated for 3 or 6 h with FGF7 (10 ng/ml) or vehicle (CTRL). RNA samples were analyzed by qRT–PCR for the indicated ISGs relative to *Rps29*.

B  Serum-starved HaCaT keratinocytes were treated for 6 h with FGF7 or FGF10 (each 10 ng/ml). RNA samples were analyzed by qRT–PCR for *IRF7* and *RSAD2* relative to *RPLP0*.

C  Serum-starved HaCaT keratinocytes were treated for 16 h with FGF7 (10 ng/ml). Protein lysates were analyzed by Western blot for IRF1, IRF3, IRF9, and GAPDH.

D  RNA samples from untreated, serum-starved primary keratinocytes from WT and *Ifnar1*-KO mice were analyzed by qRT–PCR for *Irf7*, *Rsad2*, *Oasl2*, and *Ifnar1* relative to *Rps29*.

E  Growth factor-starved primary keratinocytes from neonatal WT, *Ifnar1*-KO, or *Ifnar1/Ifnlr1*-KO mice were treated for 24 h with FGF7 (10 ng/ml). RNA samples were analyzed by qRT–PCR for *Irf7* relative to *Rps29*.

F  HaCaT keratinocytes were transiently transfected with an ISRE-Luciferase reporter construct and starved 24 h after transfection for 9 h. They were then incubated in the presence or absence of FGF7 (10 ng/ml) for 20 h. Lysates were analyzed using a DualGlo Luciferase Assay.

G, H  HaCaT keratinocytes were pre-treated for 2 h with the proteasome inhibitors MG132 (10 μm (G) or epoxomicin (100 nM) (H), followed by a 20-h treatment with FGF7 (10 ng/ml) or vehicle. RNA samples were analyzed by qRT–PCR for *IRF7* and *IRF1* relative to *RPLP0*.

Data information: Scatter plots show mean ± SEM. Mean expression levels in CTRL cell cultures (not treated with FGF7 or any inhibitor) were set to 1. In (D) and in the left panel of (E), expression levels in wt cells were set to 1, while expression levels in untreated IFN receptor knockout cells were set to 1 in the middle and right panel of (E). (A) $N = 8$–11 cultures (each from a different mouse) from four experiments. (B) $N = 3$–5 from one experiment. (C) Representative of 3 experiments. (D, E) $N = 5$–11 cultures (each from a different mouse) from two experiments. The results with *Ifnar1*-KO mice were reproduced in two additional experiments. (F) $N = 16$–18 from 4 experiments. (G) Representative of four experiments, $N = 3$ per treatment group for each experiment. (H) $N = 6$ per treatment group from two experiments. ns: non-significant, *$P \leq 0.05$, **$P \leq 0.01$, ***$P \leq 0.001$, ****$P < 0.0001$ (Mann–Whitney *U*-test (A, B, D, E, F, H) or *t*-test for assessment of FGF7 effect (G), with Welch correction for *IRF1*. Exact *P*-values are shown in Dataset EV2. Source data are available online for this figure.

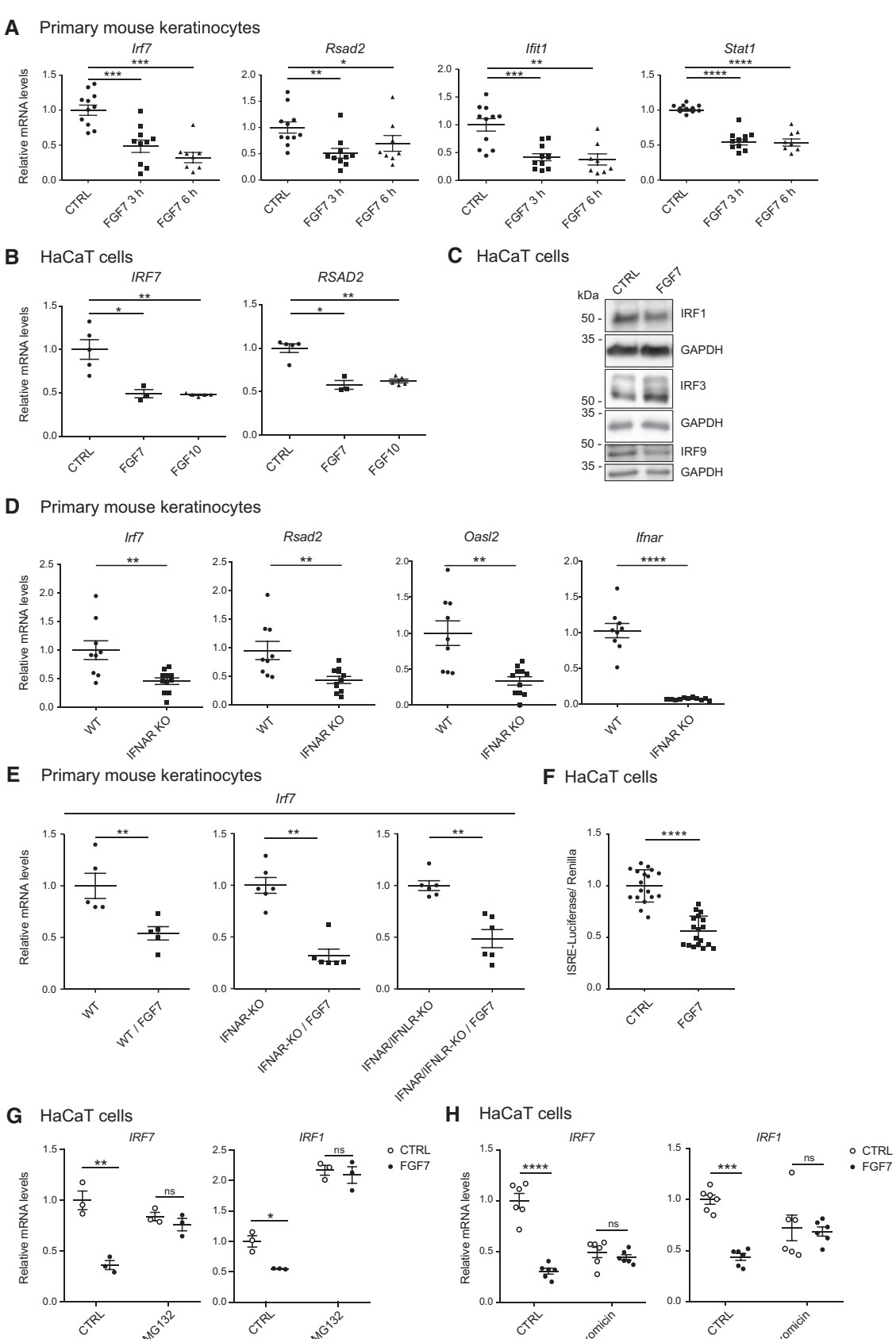

**Figure 2.**

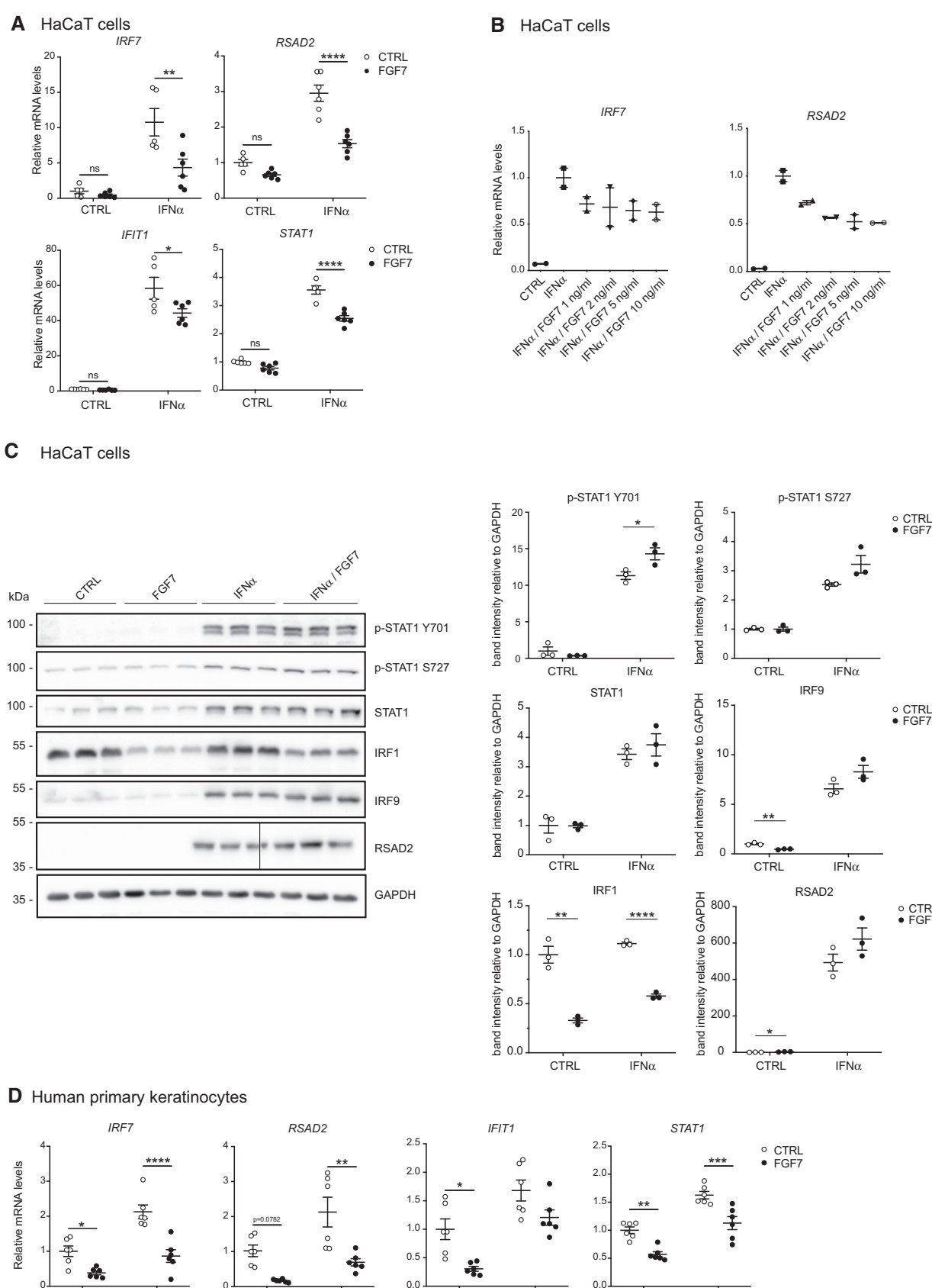

**Figure 3.**

**Figure 3. FGF7 suppresses IFN-induced ISG expression.**

A–C   Serum-starved HaCaT keratinocytes were treated with FGF7 (10 ng/ml (A, C) or different concentrations as indicated (B)) and/or IFN-α (500–1,000 U/ml) for 12–
      16 h. RNA was analyzed by qRT–PCR for ISGs relative to *RPLP0* (A, B) and protein samples by Western blot for total and pSTAT1 (Y701 and S727), IRF1, IRF9, and
      RSAD2 and GAPDH (C). The vertical line in the RSAD2 blot indicates a place where the gel was broken. Densitometric quantification of the different blots (*N* = 3) is
      shown on the right.
D     Human primary keratinocytes were starved and treated for 12 h with FGF7 (10 ng/ml) and/or IFN-α (1,000 U/ml). RNA samples were analyzed by qRT–PCR for ISGs
      relative to *RPLP0*.

Data information: Scatter plots show mean ± SEM. Mean expression levels in CTRL cell cultures were set to 1. (A) *N* = 5–6 from two independent experiments. (B)
*N* = 2–3 from one experiment. (C) Representative blot from two experiments. The intensities of the bands were used for the quantification (*N* = 3). (D) *N* = 6 replicates
from two experiments with cells from two donors. ns: non-significant, *\*P* ≤ 0.05, *\*\*P* ≤ 0.01, *\*\*\*P* ≤ 0.001, *\*\*\*\*P* < 0.0001 (2-way ANOVA with Tukey's multiple
comparisons test (A, D) or *t*-test with Welch correction for assessment of FGF7 effect (C)). Exact *P*-values are shown in Dataset EV2.
Source data are available online for this figure.

poly(I:C), which induces IFN expression by binding to toll-like receptor 3 (TLR3) (Alexopoulou *et al*, 2001), expression of *IFNB* and of *IFNL1* was strongly induced. This was accompanied by a strong increase in the expression of *IRF7, RSAD2, IRF1, CGAS, TLR3,* and *DDX58* (Fig 4A). The poly(I:C)-induced increase in the expression of *IFNB, IFNL1, IRF7, RSAD2, IRF1,* and *DDX58* was significantly suppressed by FGF7 (Fig 4A), demonstrating that FGF signaling not only interferes with the IFN response, but also with the expression of IFNs. The poly(I:C)-induced increase in TLR3 mRNA levels was also mildly, although non-significantly reduced by FGF7, while cyclic GMP-AMP synthase (*CGAS*) expression was not affected (Fig 4A). The effect of FGF7 was confirmed at the protein level for total and phosphorylated STAT1 (Y701 and S727), and for phosphorylated STAT2 (Y690), reflecting the suppression of IFN production (Fig 4B and quantification in Fig EV3). As shown before (Figs 2C and 3C), IRF1 and IRF9 expression was suppressed by FGF7 in the basal state, but this was overruled at this time point by the strong effect of poly(I:C) (Fig 4B and quantification in Fig EV3).

## FGF signaling promotes replication of Herpes Simplex Virus type 1 (HSV-1) in keratinocytes

The potent effect of FGF signaling on IFN and ISG repression raised the intriguing possibility that modulation of FGF signaling may affect viral infection. Since human keratinocytes are the first entry sites for HSV-1 (Petermann *et al*, 2015), we used this pathogen to address this question. Indeed, FGF7 boosted the production of viral glycoprotein B (*Glyc-B*) DNA and of glycoprotein D (Glyc-D) protein after HSV-1 infection of HaCaT keratinocytes, while IFN-α had the opposite effect (Fig 5A and B). 48 h after infection, HSV-1-infected cells had fused, but were still attached. In the presence of FGF7, however, most infected cells were detached (Fig 5C). The effect of FGF7 on HSV-1 viral load was dose-dependent (Fig 5D) and was also observed with human primary keratinocytes from two different donors (Fig 5E). It is

most likely mediated via the FGFR2b splice variant on keratinocytes, since the high-affinity ligands of this receptor, FGF7 and FGF10, strongly elevated Glyc-D DNA and protein levels, whereas FGF2, which binds to FGFR2b with much lower affinity (Zhang *et al*, 2006), had only a weak effect (Fig 5F). When we infected epidermal sheets from mouse tail skin *ex vivo* with HSV-1 in serum-free medium, the virus preferentially infected cells of the hair follicles as indicated by follicular staining for Glyc-D (Fig 5G, middle panel). FGF7 treatment resulted in a much larger Glyc-D-positive area, and virus-infected cells were seen throughout the interfollicular epidermis (Fig 5G, right panel). Promotion of HSV-1 infection of cultured keratinocytes by FGF7 was even observed in the presence of IFN-α (Fig 5H). There was still a considerable elevation in viral DNA levels even when FGF7 was added 4 h after addition of HSV-1 and after removal of the virus (Fig 5I). While this result does not exclude an effect on viral entry, it suggests that FGF7 mainly promotes viral replication. Consistent with this assumption, expression of nectin-1 (*NECT1*), which is required for HSV-1 entry into keratinocytes (Petermann *et al*, 2015; Sayers & Elliott, 2016), was even slightly down-regulated by FGF7 (Fig 5J).

We next treated keratinocytes with FGF7 at different time points after HSV-1 infection (Fig 6A). FGF7 suppressed ISG expression and caused a robust increase in HSV-1 DNA when added together with the virus or 2–4 h post-infection (Fig 6B and C) and was thus present for 8–12 h. However, when FGF7 was added 8 h post-infection, suppression of ISG expression and promotion of viral replication was no longer observed (Fig 6C). Importantly, Glyc-B DNA levels negatively correlated with the mRNA levels of ISGs and of IFNL2 (Fig 6B–D). This inverse correlation was also seen at the protein level: High levels of the HSV-1 Glyc-D protein correlated with reduced levels of RSAD2, IRF7, IRF9, STAT2, and IRF1, while there was no correlation with STAT1 and IRF3. In addition, Glyc-D levels inversely correlated with levels of phosphorylated STAT1 and STAT2 (Fig 6E). These findings strongly suggest that the FGF7-

**Figure 4. FGF7 suppresses poly(I:C)-induced IFN and ISG expression.**

A   Serum-starved HaCaT cells were treated for 24 h with poly(I:C) (5 μg/ml) in the presence or absence of FGF7 (10 ng/ml). RNA samples were analyzed by qRT–PCR for
    *IFNL1, IFNB, RSAD2, IRF1, IRF7, CGAS, TLR3,* or *DDX58,* relative to *RPLP0*.
B   Serum-starved HaCaT cells were treated for 18 h with poly(I:C) (5 μg/ml) in the presence or absence of FGF7 (10 ng/ml). Protein lysates were analyzed by Western
    blot for total and pSTAT1 (Y701 and S727), total and pSTAT2 (Y690), IRF1, IRF3, pIRF3 (S396), IRF9, RIG-1, RSAD2, and GAPDH.

Data information: Scatter plots show mean ± SEM. Mean expression levels in cell cultures treated with poly(I:C) only were set to 1, since IFN mRNA levels in non-
treated cells are hardly detectable. (A) *N* = 6–9 from two experiments. (B) Representative blot from two experiments. Quantification of the bands is shown in Fig EV3.
*\*\*P* ≤ 0.01, *\*\*\*P* ≤ 0.001 (2-way ANOVA with Tukey's multiple comparisons test). Exact *P*-values are shown in Dataset EV2.
Source data are available online for this figure.

**A**  HaCaT cells

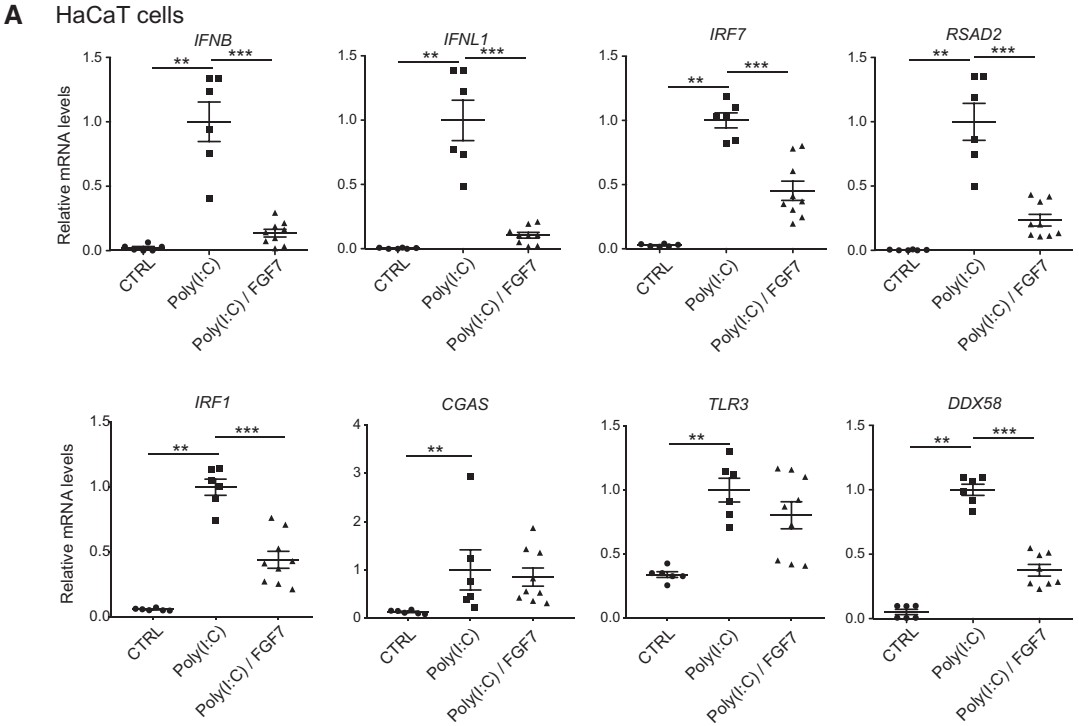

**B**  HaCaT cells

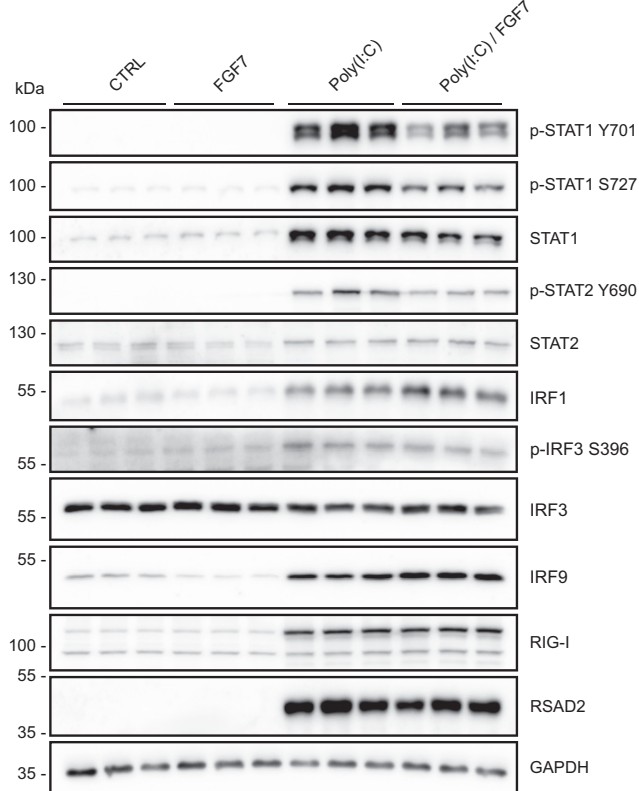

**Figure 4.**

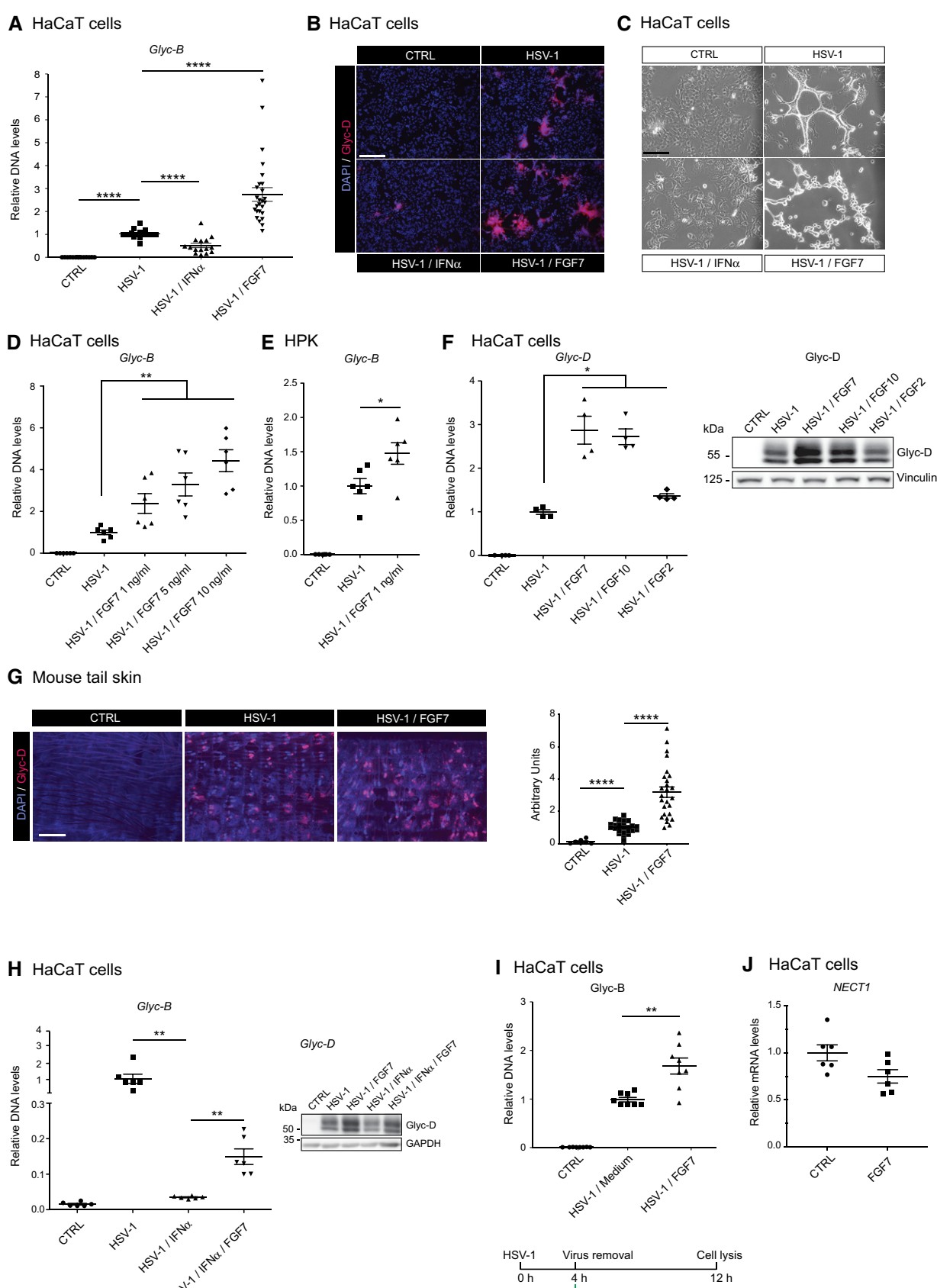

**Figure 5.**

**Figure 5. FGF7 promotes HSV-1 replication in human keratinocytes.**

A–C  Serum-starved HaCaT cells were infected with HSV-1 (MOI = 0.5) in the presence or absence of FGF7 (10 ng/ml) or IFN-α (1,000 U/ml). Viral load was determined 24 h post-infection (hpi) by qPCR for the HSV-1 *Glyc-B* gene relative to the human β-actin gene (*ACTB*) (A) or by HSV-1 Glyc-D immunofluorescence (red) and DAPI counterstaining (blue) (B). (C) Transmission microscopy images of HaCaT cells 48 h post-infection (hpi).

D, E  Serum-starved HaCaT cells (D) or human primary keratinocytes (HPK, from two donors) (E) were infected with HSV-1 (MOI = 0.5) in the presence or absence of FGF7 at the indicated concentrations. Virus load was determined by qPCR for HSV-1 *Glyc-B* relative to *ACTB* 24 hpi.

F  Serum-starved HaCaT cells were infected with HSV-1 (MOI = 0.5) in the presence or absence of FGF7, FGF10, or FGF2 (10 ng/ml). DNA samples were analyzed by qPCR for *Glyc-D* DNA relative to *ACTB*, and protein lysates were analyzed by Western blot for Glyc-D and vinculin 10 hpi.

G  Epidermal sheets from tail skin, infected *ex vivo* with HSV-1 (MOI = 2) ± FGF7 (10 ng/ml), were stained 48 hpi for Glyc-D (red) using DAPI as counterstain (blue). The infected area (red staining) was measured using ImageJ software. Data are expressed as A.U. = arbitrary unit. Background staining of hairs was subtracted manually.

H  Serum-starved HaCaT cells were infected with HSV-1 in the presence or absence of FGF7 (10 ng/ml) or IFN-α (1,000 U/ml). Viral load was determined 24 hpi by qPCR for the HSV-1 *Glyc-B* gene relative to *ACTB* or by Western blot for the viral protein Glyc-D and GAPDH.

I  Serum-starved HaCaT cells were infected with HSV-1 (MOI = 0.5). Four hours hpi, infected cells were washed and incubated in fresh serum-free medium containing FGF7 (10 ng/ml). Viral load was measured 8 h later by qPCR for *Glyc-B* relative to *ACTB*.

J  Serum-starved HaCaT keratinocytes were treated for 6 h with FGF7 (10 ng/ml). RNA samples were analyzed by qRT–PCR for *NECT1* (encoding nectin-1) relative to *RPLPO*.

Data information: Scatter plots show mean ± SEM. In (A), and (D–I) mean viral DNA levels or stained area in HSV-1-infected, but non-treated cells were set to 1. In (J), mean expression levels in CTRL cell cultures were set to 1. (A) $N = 16$–24 from four independent experiments. (B, C) Representative pictures are shown. (D) $N = 6$ from one experiment. (E) $N = 6$ from two experiments and two donors. (F) $N = 4$ from two experiments. The Western blot is a representative of two experiments. (G) $N = 16$–24 measured areas from 8 mice from 4 experiments. (H, scatter plot) $N = 6$ from two experiments. The Western blot is a representative of two experiments. (I) $N = 8$ mice from two experiments. (J) $N = 6$ from 2 experiments. *$P \leq 0.05$, **$P \leq 0.01$, ****$P \leq 0.0001$ (Mann–Whitney *U*-test). Exact *P*-values are shown in Dataset EV2. Magnification bars: 200 μm (B) or 80 μm (C) and 800 μm (G).

Source data are available online for this figure.

mediated suppression of ISG and also of IFN expression contributes to the promotion of viral replication.

## FGF signaling promotes replication of RNA viruses

The FGF7 effect was not restricted to the DNA virus HSV-1, but it also promoted replication of the RNA virus lymphocytic choriomeningitis virus (LCMV). Expression levels of LCMV nucleoprotein (NP) were strongly elevated in the presence of FGF7, while IFN-α had the opposite effect (Fig 6F).

Next, HaCaT cells were infected with two strains of the human pathogen Zika virus (ZIKV), which naturally gains access to the body via mosquito bites and can infect cells of the skin, including keratinocytes (Hamel *et al*, 2015). The number of cells expressing the ZIKV envelope protein (ZIKV-ENV) was higher in FGF7-treated

compared to mock-treated cells (Fig 6G). qRT–PCR analysis of ZIKV RNA demonstrated that FGF7 was also effective when added 2 h after infection (Fig 6H). Consistent with the important role of ISGs in the effect of FGF7, *MxA* (human myxovirus resistance protein) and *OAS1* (2′-5′-oligoadenylate synthase 1), two ISGs that are highly expressed during infection by viruses of the *Flaviviridae* family (Zhu *et al*, 2014), were upregulated during ZIKV infection, and FGF7 antagonized this effect (Fig 6I and J).

## Inhibition of FGFR signaling is an antiviral strategy

Finally, we determined whether FGFR inhibitors have antiviral activity. As expected, the FGF7-induced promotion of viral replication in serum-starved keratinocytes was suppressed by the FGFR kinase inhibitors AZD4547 and BGJ398 as determined by analysis of

**Figure 6. FGF7 suppresses virus-induced ISG expression and stimulates replication of different viruses in keratinocytes.**

A–E  Serum-starved HaCaT cells were infected with HSV-1 (MOI = 0.5) ± FGF7 (10 ng/ml), added at the indicated time points (A). 12 h after infection, RNA samples were analyzed by qRT–PCR for *RSAD2*, *IRF7*, and *IFNL2* relative to *RPLPO* (B); DNA samples were analyzed by qPCR for *Glyc-B* DNA relative to *ACTB* (C); and protein lysates were analyzed by Western blot for Glyc-D, total and phosphorylated STAT1 (Y701), STAT2, and IRF3; for IRF1, IRF7, IRF9, and RSAD2; and for GAPDH or vinculin (loading controls) (E). Quantification of the Glyc-D, RSAD2, IRF1, IRF7, IRF9, and STAT2 band intensity relative to GAPDH is shown on the right hand side. (D) shows mean *Glyc-B* DNA and RSAD2, IRF7, and IFNL2 mRNA levels (based on results shown in Fig 6B and C) in virus-infected cells treated for different periods with FGF7.

F  HaCaT cells were infected with LCMV in the presence or absence of FGF7 (10 ng/ml) or IFN-α (1,000 U/ml) and analyzed for LCMV nucleoprotein (NP) by flow cytometry 24 hpi. Arbitrary units are shown.

G, H  HaCaT cells were infected with ZIKV strains 976 (Uganda isolate) (MOI = 0.1) or PF13/251013-18 (French Polynesia isolate) (MOI = 20) and treated 2 hpi with FGF7 (20 ng/ml) or vehicle (CTRL). Immunofluorescence staining for the ZIKV envelope protein and quantification of the percentage of cells expressing the ZIKV envelope protein (G) and qRT–PCR analysis of RNA samples for ZIKV RNA (H) 48 hpi is shown.

I, J  HaCaT cells were infected with the ZIKV PF13/251013-18 (MOI = 20) in the presence or absence of FGF7 (20 ng/ml). RNA samples were analyzed by qRT–PCR for *OAS1* and *MxA* relative to *RPLPO* 72 hpi.

Data information: Scatter plots show mean ± SEM. In (B–H and J), results obtained with virus-infected, but non-treated cells were set to 1; in (I), mean expression levels in non-infected cell cultures were set to 1. (B) $N = 4$–5 from 2 experiments. (C) $N = 4$ from 2 experiments. (E): Representative blots of two experiments. (F) $N = 5$–6 from two experiments. (G, H) $N = 4$ from one experiment. (I, J) $N = 4$–8 from two experiments. Magnification bar in (G): 40 μm. *$P \leq 0.05$, **$P \leq 0.01$, ***$P \leq 0.001$ (Mann–Whitney *U*-test). Exact *P*-values are provided in Dataset EV2.

Source data are available online for this figure.

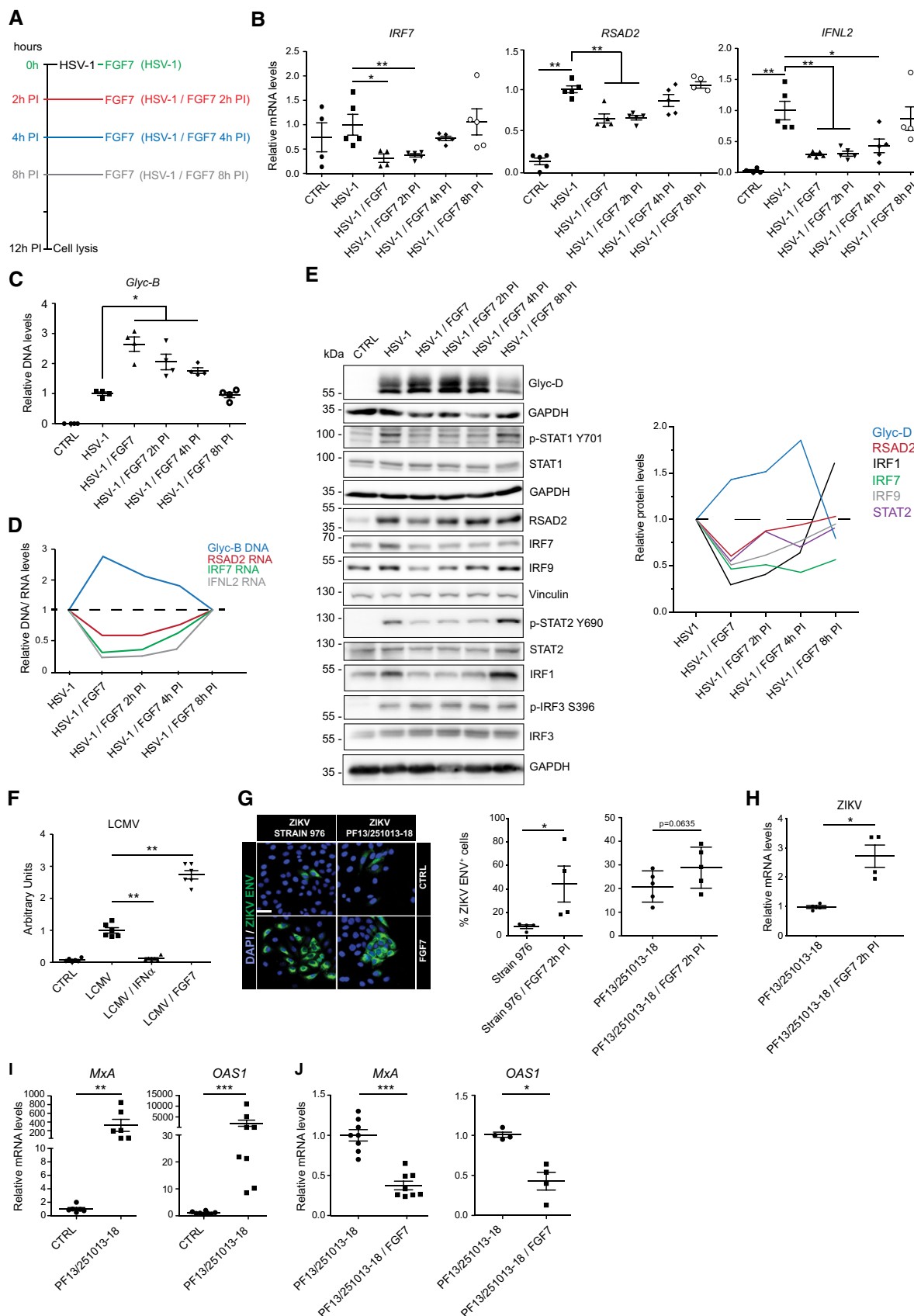

**Figure 6.**

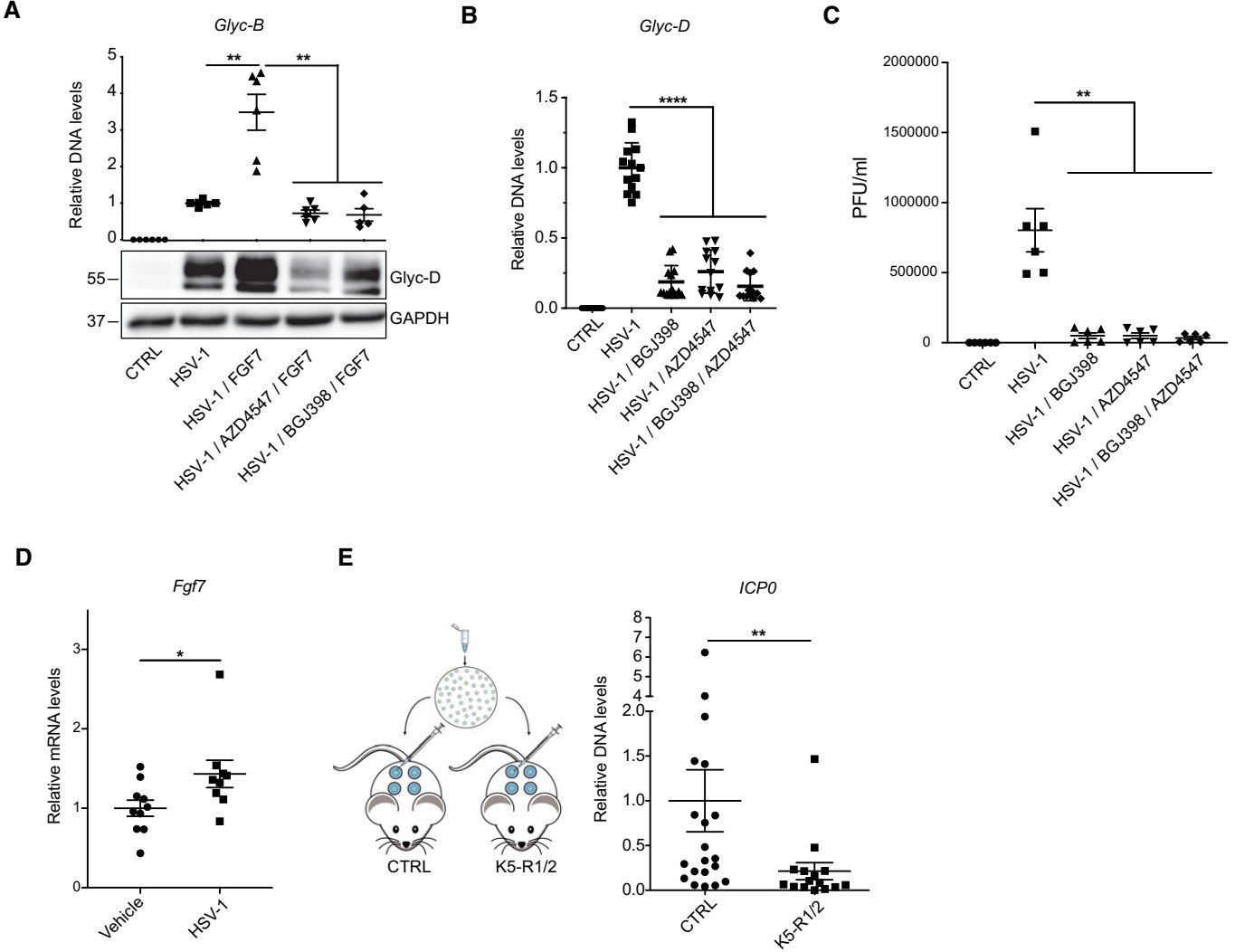

**Figure 7. Inhibition of FGFR signaling blocks HSV-1 replication.**

A    Serum-starved HaCaT cells were infected with HSV-1 (MOI = 0.5) in the presence or absence of FGF7 (10 ng/ml) and the FGFR kinase inhibitors AZD4547 (1 μM) or BGJ398 (3.5 μM). DNA and protein lysates were analyzed by qPCR for *Glyc-B* relative to *ACTB* or by Western blot for Glyc-D and GAPDH, respectively, 16 hpi.

B, C    HaCaT cells were cultured in DMEM/10% FCS (B) or DMEM/5% FCS (C) and infected with HSV-1 (MOI = 0.5) in the presence or absence of AZD45347 (1 μM) and/or BGJ398 (3.5 μM). Viral load was determined by qPCR for *Glyc-D* relative to *ACTB* (B) or by measurement of viral titers (plaque-forming units (PFU)) 14 hpi (C).

D, E    Adult C57BL/6 wild-type (D) or K5-R1/R2 mice and respective controls (E) were infected subcutaneously with HSV-1 (MOI = 10). RNA from infected skin was analyzed by qRT–PCR for *Fgf7* relative to *Rps29* (D), and DNA was analyzed by qPCR for *ICP0*, normalized to the host gene *Tbx15* 48 hpi (E).

Data information: Scatter plots show mean ± SEM. In (A, B), mean viral DNA levels in HSV-1-infected, but non-treated cells were set to 1. In (C), absolute PFU/ml cell lysate is shown. In (D), mean expression in non-infected mice (vehicle-injected) was set to 1, and in (E), mean *ICP0* DNA levels in Ctrl mice were set to 1. (A) N = 5–6 from two experiments. (B) N = 12–13 from 4 experiments. (C) N = 6 from two experiments. (D) N = 9–10 infected mouse skin spots from two experiments. (F) N = 20–24 infected mouse skin spots from two experiments. *$P \leq 0.05$, **$P \leq 0.01$, ****$P < 0.0001$ (Mann–Whitney U-test). Exact P-values are provided in Dataset EV2. Source data are available online for this figure.

*Glyc-B* DNA and Glyc-D protein levels 24 h after HSV-1 infection (Fig 7A). In the presence of serum, which contains FGFs, HSV-1 infection was effectively suppressed by treatment with these inhibitors alone or in combination as demonstrated by analysis of *Glyc-B* DNA levels or viral titers (Fig 7B and C). Since both inhibitors are highly selective for FGFR1-3, these findings strongly suggest that FGFR inhibition is sufficient to suppress viral replication in serum-containing medium, although a minor contribution of other kinases that are inefficiently inhibited by AZD4547 and BGJ398 cannot be excluded.

Finally, we examined the effects of loss of FGFR signaling on HSV-1 replication *in vivo*. 48 h after subcutaneous inoculation of HSV-1 into wild-type mice (Aoki *et al*, 2013), we observed a mild, but significantly increased expression of *Fgf7* (Fig 7D). Notably, the viral load was significantly lower in K5-R1/R2 compared with control mice 48 h after cutaneous HSV-1 infection as revealed by PCR analysis of the DNA encoding the immediate-early protein ICP0 (Fig 7E).

Taken together, these results highlight the biological relevance of the FGF-mediated regulation of ISG expression and the antiviral activity of FGFR kinase inhibition.

## Discussion

We identified an unexpected role of FGFR signaling in antiviral defense, which is mediated at least in part via control of the cellular IFN response. FGFs exert their early effect on ISG expression at the transcriptional level and in an FGFR kinase-dependent, but IFN receptor-independent manner. This conclusion is based on the findings that (i) FGF7 and FGF10 suppressed ISG expression already within 3–6 h; (ii) the FGF-mediated suppression of ISG expression was still observed in the absence of the receptors for type I and type III IFNs; (iii) expression of *SOCS1* and *SOCS3*, which encode inhibitors of the IFN receptor signaling complex, was not upregulated by FGF7; and (iv) FGF7 suppressed the expression of an ISRE reporter gene. The effect of FGF7 is likely to involve proteasomal degradation of a protein required for ISG expression, since it was abolished in the presence of different proteasome inhibitors. Our results further suggest an involvement of a combination of the MEK1/2-ERK1/2 and PI3K-AKT pathways in the FGF7-mediated ISG regulation. A role of MEK/ERK signaling is consistent with the induction of a type I IFN response in human keratinocytes by MEK inhibitors (Lulli *et al*, 2017) and with the down-regulation of the IFN-induced antiviral response by activated Ras in fibroblasts (Battcock *et al*, 2006).

The ISREs present in the promoters of ISGs are recognition sites for the IFN-stimulated gene factor 3 (ISGF3), a complex of IRF9, pSTAT1, and pSTAT2 (Darnell *et al*, 1994). ISREs also have an embedded IRF response element (IRE) (Michalska *et al*, 2018), and ISREs and/or IREs are present in the promoters of ISGs (Ivashkiv & Donlin, 2014). Through binding to ISREs and IREs, different IRFs, including IRF9, IRF1, and IRF7, regulate ISG expression (Leviyang *et al*, 2019; Panda *et al*, 2019). Since IRF and also STAT1/2 levels are frequently regulated by ubiquitination and subsequent proteasomal degradation (Nakagawa & Yokosawa, 2000; Barro & Patton, 2007; Zhao *et al*, 2016) and since inhibition of the proteasome blocked the effect of FGF7 on ISG expression, IRFs and STATs are candidate targets of the rapid effect of FGF7. Future biochemical/proteomics studies will be required to characterize the ISRE/IRE-bound proteins in the presence or absence of FGF7 or FGFR inhibitors to further elucidate the molecular mechanism of this pathway.

The two- to threefold down-regulation of the expression of various ISGs within a few hours of FGF7 stimulation is likely to be sustained, since some ISGs encode positive regulators of ISGs, such as STAT1, STAT2, and several IRFs (Darnell *et al*, 1994). In particular, the effect of FGF7 on IRF7, but also on IRF1, which was shown to regulate IFN expression (Fujita *et al*, 1989), also provides a likely explanation for the FGF-mediated impairment of IFN expression in response to poly(I:C) or viruses. This is likely to be potentiated by the FGF-mediated suppression of ISGs that encode proteins involved in pathogen sensing and subsequent induction of IFN expression.

IFNs and ISGs are known for their multiple activities in the control of cell proliferation, apoptosis, cancer inhibition, and immunomodulation (Maher *et al*, 2007; Lasfar *et al*, 2016). Thus, the results presented in this study may have widespread positive or negative consequences *in vivo*. It remains to be determined whether the increased expression of ISGs contributes to the atopic dermatitis-like phenotype that develops in K5-R1/R2 mice. This is,

however, not very likely, since ISGs were already upregulated prior to the development of the phenotype and even before the excessive infiltration of the skin by immune cells.

By contrast, the effect of FGFs on IFN and ISG expression correlated with a strong suppression of the antiviral defense, a key function of IFNs. FGF7 mainly promoted viral replication, although an effect on viral entry cannot be excluded due to the inhibitory effect of ISGs on different stages of the viral life cycle (Schneider *et al*, 2014). It has been suggested that HSV-1 enters cells via FGF receptors (Kaner *et al*, 1990). However, this cannot explain the effect that we observed, since FGF7 enhanced the viral load, while the entry of HSV-1 via an FGFR was inhibited by recombinant FGF (Kaner *et al*, 1990). In addition, our results suggest that the effect of FGFs on HSV-1 infection is not mediated via nectin-1 expression. This is further supported by the finding that FGF7 also inhibited replication of the RNA viruses ZIKV and LCMV. Rather, the effect on different viruses is consistent with the broad antiviral activities of ISGs (Schoggins & Rice, 2011).

Interestingly, FGF7 expression is upregulated in wounded/inflamed skin (Werner *et al*, 1992) and also in response to cutaneous HSV-1 infection (this study). This may promote viral replication at the site of injury/inflammation through suppression of ISG expression. Therefore, the beneficial effect of increased FGF7 expression, which includes promotion of tissue repair and protection from physical and chemical insults (Finch & Rubin, 2004; Maddaluno *et al*, 2017), may come with a risk of enhanced viral infections.

The results of our work are likely of medical relevance, since they suggest the use of FGFR inhibitors for the control of viral infections in humans. Such an approach would most likely not be confined to the skin. Consistent with this assumption, FGF2 promoted hepatitis C virion production in hepatoma cells (Van *et al*, 2016) and Zika virus infection of human fetal astrocytes (Limonta *et al*, 2019). Furthermore, FGF7 or FGF2 suppressed the expression of some ISGs in human lung epithelial cells (Prince *et al*, 2001), colon cancer cells (this study), or in human astrocytes, respectively (Limonta *et al*, 2019). However, the use of FGFR inhibitors as an antiviral strategy will require a precise timing of the treatment, since they may affect tissue healing. Indeed, inhibition of FGFR signaling impeded the repair of mouse lungs after influenza virus infection (Quantius *et al*, 2016). Besides timing, the affected cell type and its FGFR expression pattern are likely to be relevant. Thus, a subset of FGFs, which mainly activate FGFR3, inhibited infection of different types of cancer cells with vesicular stomatitis virus or Coxsackie virus through an as yet unidentified mechanism that is independent of ISG regulation (van Asten *et al*, 2018). Therefore, it will be important to determine whether certain cancer cells, among which many show deregulated FGFR signaling (Tanner & Grose, 2016), may not be protected from viruses by FGFR inhibitors, but rather become more susceptible. Furthermore, other effects of FGFs on the viral life cycle need to be considered as recently shown for Dengue virus, where inhibition of FGFR4 decreased viral replication, but increased the infectivity of the resulting virions (Cortese *et al*, 2019).

In spite of these open questions, the prospect of using FGFR inhibition as an antiviral strategy is promising, since FGFR kinase inhibitors, FGFR neutralizing antibodies, or FGF ligand traps are in clinical trials for the treatment of different types of cancers and are generally well tolerated (Touat *et al*, 2015; Tanner &

Grose, 2016). Obviously, such a novel indication for FGFR inhibitors will require intensive studies regarding dosing, mode of application, efficacy against different viruses, and side effects in virus-infected patients. Nevertheless, the effort seems to be justified given the limited options for the treatment of viral infections, of which some have a high mortality rate or for which vaccines are not yet available, such as SARS-CoV-2. Virostatic agents, which interfere with the viral life cycle, are frequently employed. However, they are most often virus-specific and thus susceptible to viral variation, and they often exhibit high toxicity. Thus, improved strategies are urgently needed, and FGFR inhibition is a fundamentally different approach.

# Materials and Methods

### Antibodies, recombinant proteins, and chemical inhibitors

The following recombinant proteins and chemical inhibitors were used: Human FGF7 (100-19, PeproTech Inc., Rocky Hill, NJ or R&D Systems, Minneapolis, MN), human FGF10 (100-26, PeproTech Inc.), human FGF2 (100-18, PeproTech Inc.), human IFN-α (300-02AA, PeproTech Inc), FGFR1/2/3 inhibitors AZD4547 (S2801, Selleckchem, Houston, TX) and BGJ398 (NVP-BGJ398; S2183, Selleckchem), PI3K inhibitor LY294002 (InvivoGen, San Diego, CA), AKT1/2 inhibitor A6730 (A6730, Sigma, Munich, Germany), Mek1/2 inhibitor U0126 (662005, Calbiochem, San Diego, CA), PLC-γ inhibitor U73122 (1278, Tocris, Bristol, UK), and proteasome inhibitors MG132 (S2619 Selleckchem), bortezomib (PS-341) (S1013, Selleckchem), and epoxomicin (E3652, Sigma).

### Genetically modified mice and infection of mice with HSV-1

K5-R1/R2 and control mice (mice with floxed *fgfr1/fgfr2* alleles, but without Cre, or K5-Cre mice with wild-type *fgfr* alleles) in C57BL/6 genetic background and mice lacking IFNAR1 or both IFNAR1 and IFNLR1 in C57BL/6 genetic background were described previously (Ank *et al*, 2008; Mordstein *et al*, 2008; Yang *et al*, 2010). They were maintained under specific pathogen-free conditions and received food and water *ad libitum*. For HSV-1 infection, female mice at the age of 8–10 weeks were shaved, and 24 h later, each mouse received four subcutaneous injections of 50 μl HSV-1 (MOI = 10) on the flank. 48 h later, they were sacrificed and the amount of *ICP0* DNA in the injected skin was determined using primers 5′-ATA AGT TAG CCC TGG CCC CGA-3′ and 5′-GCT GCG TCT CGC TCC G-3′. *ICP0* DNA levels were normalized to the host *Tbx15* DNA (primers: 5′-TCC CCC TTC TCT TGT GTC AG-3′ and 5′-CGG AAG CAA GTC TCA GAT CC-3′) (Mohn *et al*, 2009). For the establishment of primary keratinocytes, we used neonatal mice at P3 from both genders. Mouse maintenance and all animal experiments were performed according to Swiss Law and approved after in-depth review and approval by the local veterinary authorities of Zurich, Switzerland (Kantonales Veterinäramt Zürich).

### Separation of dermis from epidermis of mouse back skin

Separation of epidermis from dermis was achieved either by heat shock treatment (30 s at 55–60°C followed by 1 min at 4°C, both in PBS), by incubation for 50–60 min at 37°C in 0.143% dispase (17105-041, Life Technologies, Carlsbad, CA) in DMEM or by incubation in 0.8% trypsin (27250-018, Life Technologies)/DMEM for 15–30 min at 37°C. For dispase and trypsin treatment, the subcutaneous fat was gently scraped off with a scalpel prior to incubation.

### Immunofluorescence analysis of skin sections and cultured cells

Frozen mouse back skin sections were fixed with cold methanol, and unspecific binding sites were blocked with PBS containing 2% bovine serum albumin (BSA) (Sigma)/1% fish skin gelatin (Sigma)/0.05% Triton X-100 (Carl Roth GmbH, Karlsruhe, Germany) for 2 h at room temperature. Samples were then incubated overnight at 4°C with the primary antibodies diluted in the same buffer. After three washes with PBS-T [1 × PBS/0.1% Tween-20 (Carl Roth GmbH)], slides were incubated at room temperature for 4 h with secondary antibodies and DAPI (4′,6-diamidino-2-phenylindole dihydrochloride) (Sigma) as counterstain and washed with PBS-T again and mounted with Mowiol (Hoechst, Frankfurt, Germany). Stained sections were photographed using a Leica SP1-2 confocal microscope equipped with a 63 × 0.6–1.32 NA (Iris) PL Apo Oil objective. Data acquisition was performed using Leica Confocal Software (Leica, Wetzlar, Germany). For immunofluorescence analysis of cultured cells, cells were washed in PBS and then either fixed for 5 min with cold methanol for staining with antibodies against ZO-1 and IRF7, or with 4% paraformaldehyde (PFA) (Sigma) for 20 min at RT for Glyc-D staining. PFA-fixed cells were then incubated for 10 min with 0.5% Triton X-100 in PBS. After 1 h blocking in PBS containing 2% BSA, cells were stained with primary antibodies for 1 h in the same blocking buffer. After three washes with PBS, cells were incubated with secondary antibodies and DAPI. Stained cells were photographed using a Zeiss Imager.A1 microscope equipped with an Axiocam MRm camera and EC Plan-Neofluar objectives (10×/0.3, 20×/0.5). For data acquisition, we used the Axiovision 4.6 software (all from Carl Zeiss Inc., Jena, Germany). The Glyc-D positive area was quantified by digital pixel measurements using ImageJ software. Antibodies used for immunofluorescence staining are listed in Appendix Table S1.

### RNA isolation and qRT–PCR

Total RNA from the epidermis of mice or from total skin was isolated with Trizol followed by purification with the RNeasy Mini Kit, including on-column DNase treatment (Qiagen, Hilden, Germany). Total RNA from cultured cells was extracted directly with the RNeasy Mini Kit. cDNA was synthesized using the iScript kit (Bio-Rad Laboratories, Berkeley, CA). Relative gene expression was determined using the Roche LightCycler 480 SYBR Green system (Roche, Rotkreuz, Switzerland). Expression of the genes encoding ribosomal protein L29 (*Rpl29*) or ribosomal protein lateral stalk subunit P0 (*RPLP0*) was used for normalization of expression levels of murine or human genes, respectively, unless otherwise indicated in the figure legends. Primers used for qRT–PCR are listed in Appendix Table S2.

### Gene expression profiling and bioinformatic analysis

Epidermis from nine K5-R1/R2 and nine control mice at P18 was separated from the dermis and immediately frozen in liquid nitrogen. Samples from three mice per genotype were pooled and

subjected to Affymetrix microarray hybridization ($N = 3$ pools per genotype) (Beyer et al, 2008). Genes, which were significantly and more than 1.5 $\log_2$ (fold change) regulated in K5-R1/R2 compared with control mice, were entered into EnrichR to discover functional enrichments of the FGF-regulated genes (Kuleshov et al, 2016). Tables referring to Gene Ontology (GO) Biological Process 2017b and Reactome 2016 were downloaded and formatted into tables. Lists of significantly upregulated genes, ISGs, and original tables from functional enrichments can be found in Dataset EV1.

## Cell culture

Primary keratinocytes were isolated from neonatal mice (Yang et al, 2010) and cultured for 3 days in defined keratinocyte serum-free medium (dK-SFM) (Invitrogen, Carlsbad, CA) supplemented with 10 ng/ml EGF, $10^{-10}$ M cholera toxin, and 100 U/ml penicillin/100 μg/ml streptomycin (Sigma). Plates were coated with collagen IV (Sigma) prior to seeding the cells.

Human primary foreskin keratinocytes were established from foreskin (Strittmatter et al, 2016). Foreskin samples had been obtained upon informed written consent of the parents and upon approval by the local ethical committee (KEK-ZH-Nr. 2015-0198) and according to the Declaration of Helsinki Principles. They were cultured in CnT-Prime Epithelial Cell Culture Medium (CELLnTEC, Bern, Switzerland) and used for experiments between passages 3 and 5.

HaCaT keratinocytes (Boukamp et al, 1988) were obtained from the owner, Prof. P. Boukamp, German Cancer Research Center Heidelberg, and Caco-2 colon cancer cells were from Sigma. Both cell lines were cultured in DMEM (Sigma) supplemented with 10% FCS (Thermo Fisher Scientific, Waltham, MA).

For treatment of cells with FGF7, FGF10, FGF2 (Peprotech; 10 ng/ml, unless otherwise indicated in the legends), IFN-α (Peprotech, 500–1,000 U/ml), or poly(I:C) (Sigma, 5 μg/ml, unless otherwise indicated in the legends), cells were grown to confluency, starved overnight in serum-free medium, and treated for different time periods. In some experiments, cells were pre-incubated with different inhibitors or vehicle (DMSO) for 2–3 h at 37°C, prior to treatment with FGF7.

## Cell transfection and luciferase assay

Transient transfection of HaCaT cells was performed with a TK-Renilla luciferase expression vector and the pGL4.45[luc2P/ISRE/Hygro] reporter vector (Promega, Fitchburg, WI), the latter containing five copies of an ISRE that drives transcription of the luciferase reporter gene. Cells were seeded into 24-well plates, cultured for 24 h, and transfected using Xfect™ (Takara, Kusatsu, Japan). Subsequently, cells were starved in serum-free medium overnight, then treated with FGF7 (10 ng/ml) for 20 h, lysed, and assayed with a dual-luciferase assay system (Promega) according to the manufacturer's instructions. Relative light units were measured in a GloMax Discover Microplate (Promega). Firefly luciferase activity was normalized to Renilla luciferase activity (transfection control).

## Preparation of protein lysates and Western blot

Cells were harvested in T-PER tissue protein extraction reagent (Pierce, Rockford, IL) containing Complete Protease Inhibitor Cocktail (Roche). Lysates were cleared by centrifugation (16,000 $g$, 30 min, 4°C), snap-frozen, and stored at $-80°C$. The protein concentration was determined using the BCA Protein assay (Pierce). Proteins were separated by SDS–PAGE and transferred onto nitrocellulose membranes. Membranes were incubated with primary antibodies and, subsequently, antibody-bound proteins were detected with horseradish peroxidase-coupled antibodies against goat-IgG (Sigma), rabbit-IgG, or mouse IgG (all from Promega). Antibodies used for Western blot analysis are listed in Appendix Table S1.

## HSV-1 production and cell infection

HSV-1 was produced as described (Strittmatter et al, 2016). Subconfluent keratinocytes were starved overnight in serum-free medium and then incubated with HSV-1 (MOI = 0.5) in the presence or absence of FGF7, FGF10, or FGF2. Where indicated, infection was preceded by treatment with AZD4547 (Selleckchem, 1 μM), BGJ398 (Selleckchem, 3.5 μM), or DMSO (vehicle) for 3 h before treatment with FGF7. Infected cells were then cultured for 10-16 h before the assessment of viral load. Alternatively, subconfluent HaCaT cells were infected in DMEM/5% FCS with HSV-1 (MOI = 0.5) in the presence or absence of AZD4547 and/or BGJ398 and used for the analysis of viral load 14 h later.

## Preparation of genomic and viral DNA from HSV-1-infected cells

Genomic and viral DNA were obtained from infected cells as previously described (Strittmatter et al, 2016). Samples were used for quantitative PCR to measure HSV-1 virus load. Primers for amplification of the human β-actin DNA (5′-TAC TCC TGC TTG CTG ATC CAC-3′ and 5′-TGT GTG GGG AGC TGT CAC AT-3′) and the viral glycoprotein B (Glyc-B) DNA (5′-CGC ATC AAG ACC ACC TCC TC-3′ and 5′-GCT CGC ACC ACG CGA-3′) were used.

## Analysis of HSV-1 viral titers

Viral titers were measured using plaque assay. Infected cells were scraped in 500 μl DMEM and lysed by single freeze-thawing. 24-well plates with Vero cells were infected at 37°C for 2 h with serial 10-fold dilutions of the cell lysate. After infection, the lysate was removed and the cells were washed twice with PBS and overlaid with DMEM containing 0.7% agarose. Two days later, the cells were fixed and stained overnight with a solution containing 0.2% crystal violet, 11% formaldehyde, 2% ethanol, and 2% paraformaldehyde. The next day, the plates were washed with water. Plaques were counted and plaque forming units (PFUs) were calculated taking into consideration the dilution factor.

## Ex vivo HSV-1 infection

HSV-1 infection of mouse tail epidermis was previously described (Rahn et al, 2015). Briefly, skin was prepared from the tails of 3-month-old mice. The epidermis was separated from the dermis by dispase treatment (5 mg/ml). After floating the epidermal sheets on serum-free DMEM overnight, they were incubated with HSV-1 alone (MOI = 2) or in combination with FGF7 (10 ng/ml) for 48 h and subsequently fixed in 4% PFA for 1 h at RT. Tissues were incubated in blocking solution (PBS 2% BSA/1% fish skin gelatin/0.05%

Triton X-100) for 2 h at RT and stained overnight at 4°C with an anti-Glyc-D antibody. After three washes in PBS, epidermal sheets were incubated for 4 h with AF555-conjugated anti-mouse IgG and DAPI diluted in PBS/0.05% Triton X-100 at RT and mounted with their basal side on top of a specimen slide using Mowiol.

### LCMV infection and flow cytometric analysis

Subconfluent HaCaT cells were serum-starved overnight and incubated overnight at 37°C with LCMV (MOI 0.05 and 0.2). Cells were detached from the 12-well plate using 1% trypsin and fixed/permeabilized in 500 μl 2 × FACS Lyse (Becton Dickinson, Franklin Lakes, NJ) with 0.05% Tween-20 for 10 min at room temperature. After washing, intracellular staining was performed for 30 min at room temperature using the LCMV nucleoprotein-specific antibody VL-4. After washing, they were resuspended in PBS/1% PFA. Flow cytometric analysis was performed using an LSRII flow cytometer (Becton Dickinson). Raw data were analyzed using FlowJo software (Tree Star Inc, Ashland, OR).

### ZIKV infection

HaCaT cells were seeded on a four-well tissue chamber on a PCA slide ($5 \times 10^4$/chamber). After overnight serum starvation, cells were infected with the ZIKV strains Uganda (strain 976) (MOI = 0.1) (kindly provided by the European Virus Archive goes Global (EVAg) project) or French Polynesia (PF13/251013-18) (MOI = 20). Two hours post-infection, cells were treated with 20 ng/ml FGF7 or left untreated. Culture medium ± FGF7 was changed every day. 48 h post-infection, cells were either analyzed by immunofluorescence using a Flavivirus group-specific antibody (4G2) detecting the ZIKV envelope protein (ZIKV-Env) or harvested and analyzed for ZIKV expression levels by qRT–PCR relative to human *RPL27* (ZIKV primers: 5′-AGA TCC CGG CTG AAA CAC TG-3′; 5′-TTG CAA GGT CCA TCT GTC CC-3′).

Alternatively, HaCaT cells were seeded in six-well plates. After overnight serum starvation, confluent cells were infected with the ZIKV strain French Polynesia (PF13/251013-18) (MOI = 20) in the presence or absence of FGF7 (20 ng/ml). 72 h post-infection, cells were analyzed for *OAS1* and *MxA* expression levels by qRT–PCR relative to *RPL27*.

### Statistical analysis

For all experiments, the sample size was estimated based on previous experience. There was no exclusion of any animal or data point. Analysis of stained skin sections or epidermal sheets was performed blinded by the investigators. Group allocation of the mutant mice could not be performed blinded, because the phenotype of the K5-R1/R2 mice is obvious macroscopically and histologically. Experiments were performed at least twice, and the number of replicate experiments is indicated in the figure legends.

Statistical analysis was performed using the PRISM software (Graph Pad Software Inc., San Diego, CA). Mann–Whitney *U*-test for non-Gaussian distribution or *t*-test with Welsh correction was used for experiments examining differences between two groups. For comparison of more than two groups, we used Brown-

---

### The paper explained

#### Problem
Viral infections are a major threat to human health. Combating such infections and prevention of a pandemic requires the development of vaccines, but also of efficient therapeutics. Unfortunately, many therapeutics are virus-specific and/or have high toxicity. Therefore, the development of novel approaches for the treatment of viral infections is essential.

#### Results
We identified a previously uncharacterized role of fibroblast growth factors (FGFs) in antiviral defense. FGFs efficiently suppress the interferon response and thus the expression of major antiviral genes and proteins under basal conditions and in response to interferon treatment or viral infections, while FGF receptor inhibition has the opposite effect.

#### Impact
Our study suggests the use of FGF receptor inhibitors for the treatment of viral infections. This is a promising approach, since such inhibitors are already in clinical trials for the treatment of different types of cancers and are relatively well tolerated. Due to the widespread activities of FGFs and interferons, it will also be important to study the relevance of their unexpected cross-talk described in this study in various other human diseases as well as in tissue repair.

---

Forsythe test to verify equal variance between groups, followed by two-way ANOVA and Tukey's multiple comparisons post-hoc tests.

## Data availability

The datasets produced in this study are available in the following databases: Microarray data: Gene Expression Omnibus GSE111274 (https://www.ncbi.nlm.nih.gov/geo/query/acc.cgi?acc=GSE111274).

**Expanded View** for this article is available online.

## Acknowledgements

We thank Stefanie Trautweiler, ETH Zurich, for experimental help, Drs. Andrii Kuklin, ETH Zurich, Gian Paolo Dotto, University of Lausanne, Cord Brakebusch, University of Copenhagen and Dagmar Moersdorf, University of Cologne, for helpful suggestions, Drs. Juha Partanen, University of Helsinki, and David Ornitz, Washington University St. Louis, for the *Fgfr1* and *Fgfr2* floxed mice, respectively, Dr. Petra Boukamp, German Cancer Research Center Heidelberg, for early passage HaCaT keratinocytes, and Dr. Didier Musso, Research and Diagnosis Laboratory, Institute Louis Marlade, Papeete, Tahiti, for ZIKV Polynesia strain PF 13/251013-18. This work was supported by the Swiss National Science Foundation (grants 31003A_169204 and 31003B-189364 to S.W), the ETH Zurich (grant ETH-06 15-1 to S.W. and L.M.), a Marie Curie postdoctoral fellowship from the European Union (to L. M.), a Swiss Government Excellence Postdoctoral Scholarship (to K. N.), and the European Union Horizon 2020 research and innovation program (grant agreement No 653316; "European Virus Archive goes Global (EVAg) project") that provided the ZIKV strain Uganda (strain 976).

## Author contributions

LM designed the study together with SW. LM, CU, TR, MM, LF, DSto, RS, DSte, DB, NL, GS, KN, and H-DB performed experiments and analyzed data; MW performed the bioinformatics analysis; LM and CU made the figures; PS provided IFN receptor knockout mice; EH supervised the Zika virus experiments; and AO supervised the LCMV and *in vivo* HSV-1 experiments. SW designed the study together with LM, supervised the work and provided the funding. All authors made important comments to the manuscript.

## Conflict of interest

L.M., M.M., and S.W. have filed a patent for the use of FGFR inhibitors in antiviral defense (Filing No. EP3308786A1). There are no additional competing interests of any of the authors.

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
