## [Review Process File · EMBO Molecular Medicine]

Antagonism of interferon signaling by fibroblast growth factors promotes viral replication

Luigi Maddaluno, Corinne Urwyler, Theresa Rauschendorfer, Michael Meyer, Debora Stefanova, Roman Spoerri, Mateusz Wietecha, Luca Ferrarese, Diana Stoycheva, Daniela Bender, Nick Li, Gerhard Strittmatter, Khondokar Nasirujjaman, Hans-Dietmar Beer, Peter Staeheli, Eberhard Hildt, Annette Oxenius and Sabine Werner

DOI: 10.15252/emmm.201911793

Corresponding author(s): Sabine Werner (sabine.werner@biol.ethz.ch) , Luigi Maddaluno (luigi.maddaluno@gmail.com)

Review Timeline:

Submission Date:	20th Nov 19
Editorial Decision:	20th Dec 19
Revision Received:	27th Apr 20
Editorial Decision:	16th Jun 20
Revision Received:	27th Jun 20
Accepted:	30th Jun 20

Editor: Celine Carret

Transaction Report:

20th Dec 2019

Dear Sabine,

Thank you for the submission of your manuscript to EMBO Molecular Medicine. We have now heard back from the three referees whom we asked to evaluate your manuscript.

You will see below that the three referees find the study interesting and well done. While ref. #1 is enthusiastic and only requests supportive experiments, quantitation, more controls and explanations about statistics and sampling, ref. #2 and #3 share a similar concern and would like to see more mechanism. In addition ref. #2 asked whether the data can be applied to other cell types. Overall, we agree that all these requests would make the study stronger and I'd like to invite you to address those comments in a major revision of your work.

We would therefore welcome the submission of a revised version within three to six months for further consideration and would like to encourage you to address all the criticisms raised as suggested to improve conclusiveness and clarity. Please note that EMBO Molecular Medicine strongly supports a single round of revision and that, as acceptance or rejection of the manuscript will depend on another round of review, your responses should be as complete as possible.

I look forward to receiving your revised manuscript.

Happy Holidays!

Celine Carret

Celine Carret, PhD
Senior Editor
EMBO Molecular Medicin

***** Reviewer's comments *****

Referee #1 (Comments on Novelty/Model System for Author):

the authors have confirmed the results in the human keratinocyte cell line HaCaT and have also conducted individual experiments in primary keratinocytes. In addition, functional assays were performed to confirm the biological relevance of their data.

Referee #1 (Remarks for Author):

In the present study Maddaluno et al. investigated the effect of FGF on interferon signaling in keratinocytes. For this purpose, mice without FGFR1, FGFR2 or both receptors in keratinocytes of the epidermis and hair follicles were used to identify FGF targets in the epidermis. Further analyses included the genetic or pharmacological inhibition of FGF receptors in primary mouse keratinocytes or the human keratinocyte cell line HaCaT. In addition, the results in primary human keratinocytes were partially confirmed. They found that inhibition of the FGF receptor promotes expression of interferon-stimulated genes (ISG) and proteins in vitro and in vivo. In accordance with these results FGF7 or FGF10 treatment of keratinocytes suppressed ISG expression both basally and after IFN or Poly(I:C) treatment. Further, by infection studies of keratinocytes with HSV-1, ZIKV, or lymphocytic choriomeningitis virus they demonstrated that the effect of FGF on ISG is of biological relevance: Stimulation with FGF increased the expression of genes of the virus, while inhibition of FGFR signaling blocked HSV-1 replication in cultured human keratinocytes and mice.

The study is well conducted and clearly written. In addition, it is also of interest to more than one professional audience. Nevertheless, there are a few minor points that the authors should address:

1. The panel of IFN-stimulated genes whose expression was analyzed in this study should be more consistent between the different but related experiments to facilitate comparison. For example, in Fig. 1H HaCaTs were analyzed for ISG15 expression, but not for IRF7 and Oasl2, whose expression was previously shown in mouse analysis. Moreover in Fig 2B HaCaTs were analyzed for IRF1, IRF9 and SOCS3 but therefor no data was shown in mice whereas mice were analysed for STAT2 and Irf1 which was not the case in HaCaTs.

Please also use the same please also use the same order of presentation in your graphics, this will make it easier to read and understand the graphics (in some cases this has already been done).

2. Figure S2: Inhibition of Erk1/2, PI3K or Akt does not affect the FGF7-mediated suppression of ISG expression, while inhibition of proteasomal activity abrogates the FGF7 effect. Please insert a positive control for the inhibitor so that you can be sure that the inhibitor itself works.

3. Westernblots: it's not clear from the legend how high the n-number is. Means one experiment =n=1 or one experiment with n=?.

In order to increase the significance of the western blot, a densitometric evaluation of all westernblots should be carried out and their evaluation should be presented as a graph in addition to the existing exemplary image.

a. Fig. 3C the differences in pSTAT1 activation between IFNa and INFa+FGF7 are hardly to see, this may be improved by densitometric analysis (see above).

b. Fig. 4B the western blot for IRF3 and pIRF3 is not convincing . The manuscript also mentions

STAT3, which is not shown in the figure.

c. Fig. 6E the western blot is unfortunately not convincing because of the GapDH levels, which are much lower in the FGF7 stimulated cells than in the KCtrl or HSV-1 cells.

4. Fig 6G und follows: please explain the legend of the figure. Stain 976 of ZIKV? PF13/251013-18?

5. Fig7: Please control Fig 7. Some of the graphs are shifted.

6. Diskussion: Nectin-1 was shown to be involved in HSV-1 entry in keratinocytes (Charlotte L. Sayers and Gillian Elliott 2016). Is the expression of nectin-1 influenced by FGF treatment?

7. Human keratinocytes: Mean n=6 that keratinocytes from 6 different donors were investigated or are they replicates? Please indicate it clearly in the legend

8. Statistical analysis: which test was used to test if the values come from a Gaussian distribution? Two-way ANOVA assumes that your replicates are sampled from Gaussian distributions, especially with small sample sizes this is important and should be taken into account.

Referee #2 (Remarks for Author):

Maddaluno and colleagues report the very interesting, important and original observation that signaling through FGFR1 and FGFR2 inhibits interferon stimulated gene (ISG) expression. They further show that this is functionally relevant in the context of viral infection. They propose that FGFR inhibitors could be used as broadly active antivirals.

The experiments are very well done, and the results support the conclusions.

I have two major considerations:

1. The big missing part of this exciting story is the mechanism. How does FGF7 inhibit both tonic and induced ISG expression?

The authors provide some insights with the proteasome inhibitor experiments (Figures 2G, 2H, S2G). They speculate that "The effect of FGF7 is likely to involve proteasomal degradation of a protein required for ISG expression, since it was abolished in the presence of different proteasome inhibitors." They further speculate that this mechanism could involve the well known ISG transcriptional regulators IRF9, IRF1, IRF7 and STAT1 and STAT2.

There is indeed a slight reduction of the protein levels of this transcription factors (shown in figures 2C, 3C, and 6E) but in parallel there is a reduction of the corresponding mRNAs that could very well explain the reduced protein levels. The experiments with the proteasome inhibitor provide only circumstantial evidence. The authors should further explore this. What is known about regulation of the proteasome pathway by FGF7? Is the presumed FGF7 induced proteasome activation non-specific, or specifically/preferentially targeting ISG transcriptional regulators such as IRF7 and IRF1? What about mTORC1, a central regulator of proteasome assembly (and a potential target of FGF signaling downstream of Akt)?

The authors also provide some mechanistic insights by showing that blockade of the major FGFR signaling pathways, the PI3-kinase/Akt and Erk1/2 pathways, do not suppress the effect of FGF7 on ISG expression. They conclude that "other pathways or pathway combinations are involved." What about PLCy or STATs? Is the FGFR kinase activity required the observed effects on ISG expression?

2. How generally valid is this observation? The authors focus their study on keratinocytes. What about other cells in the body? This seems an important question, since the authors propose FGFR inhibition as an exciting new antiviral strategies, also for viruses that do not only replicate in keratinocytes.

Specific points:

1. SOCS1 rather than SOCS3 is the canonical inhibitor of IFNAR signaling. Is SOCS1 regulated by FGFR2?

2. In many figures, qPCR values are shown as relative mRNA levels. Relative to what? Please indicate in each case in the figure legend or the figure.

3. There is also another concern about showing these relative mRNA levels. For example in Figure 2D, *Irf7* mRNA is relatively lower in IFNAR KO compared to wildtype. In Figure 2E, untreated wild type and IFNAR KO have the same mRNA level of *Irf7* (around 1), and in both cases it goes down after FGF7 treatment. Showing all results in one figure would probably be preferable to avoid apparent inconsistencies between Fig 2D and 2E.

4. The figure legends are often un-sufficient. Length of treatments and concentration of reagents are often missing. Y axis description often missing. Authors should please go carefully through figures and legends.

5. The panels of Figure 7 are partially overlapping and seem to be dislocated.

Referee #3 (Comments on Novelty/Model System for Author):

frankly i am not too confident about the therapeutic applications of these findings

Referee #3 (Remarks for Author):

This MS reports the finding that Mice with a KO of FGF receptors (FGFR) 1 and 2 in their skin have high levels of expression of IFN stimulated genes (ISG). Further investigation of this initial observation leads to the conclusion that FGF signaling inhibits the expression of ISGs in mouse and human keratinocytes and, as such, promotes the replication of HSV1 and other viruses in the same cells. This notion is supported by a variety of experiments using FGF7 treatment of Keratinocytes, by showing that ISG expression is stimulated by FGFR inhibitors, and that FGF signaling competes with IFN or Poly:IC treatment in regulating ISG expression. Finally it is shown that the replication of HSV1, LMC and zika viruses is promoted by FGF signaling by dampening the IFN response and that inhibition of FGF signaling adversely affects HSV 1 replication.

The results presented here are quite convincing and the major conclusion of the MS, i.e. that FGF signaling antagonizes IFN signals is amply documented.

The major weakness of this report is however that it does not identify or even investigate the mechanisms responsible for the phenomenon described. Furthermore I noticed that, in most

experiments testing the expression of the ISGs in response to FGF treatment (figs. 1,2,3,4, 6), the effect on the RNA levels of the genes are much stronger than the effect on protein levels, that in some cases show no difference. Unfortunately the genes whose expression is determined at the RNA level and those whose protein expression is shown are often not the same, making difficult to ascertain how general is this phenomenon. This discrepancy has to be discussed or explained if this MS has to be accepted for publication.

Two minor points: 1) it would be better to measure viral titers, in Infectious units, rather than DNA levels.

2)The Ms is extremely long and could be trimmed considerably. The discussion is rambling, never discusses mechanisms and some of the therapeutic implications could be toned down.

Referee #1 (Comments on Novelty/Model System for Author):

The authors have confirmed the results in the human keratinocyte cell line HaCaT and have also conducted individual experiments in primary keratinocytes. In addition, functional assays were performed to confirm the biological relevance of their data.

Referee #1 (Remarks for Author):

In the present study Maddaluno et al. investigated the effect of FGF on interferon signaling in keratinocytes. For this purpose, mice without FGFR1, FGFR2 or both receptors in keratinocytes of the epidermis and hair follicles were used to identify FGF targets in the epidermis. Further analyses included the genetic or pharmacological inhibition of FGF receptors in primary mouse keratinocytes or the human keratinocyte cell line HaCaT. In addition, the results in primary human keratinocytes were partially confirmed. They found that inhibition of the FGF receptor promotes expression of interferon-stimulated genes (ISG) and proteins in vitro and in vivo. In accordance with these results FGF7 or FGF10 treatment of keratinocytes suppressed ISG expression both basally and after IFN or Poly(I:C) treatment. Further, by infection studies of keratinocytes with HSV-1, ZIKV, or lymphocytic choriomeningitis virus they demonstrated that the effect of FGF on ISG is of biological relevance: Stimulation with FGF increased the expression of genes of the virus, while inhibition of FGFR signaling blocked HSV-1 replication in cultured human keratinocytes and mice.

The study is well conducted and clearly written. In addition, it is also of interest to more than one professional audience. Nevertheless, there are a few minor points that the authors should address:

1. The panel of IFN-stimulated genes whose expression was analyzed in this study should be more consistent between the different but related experiments to facilitate comparison. For example, in Fig. 1H HaCaTs were analyzed for ISG15 expression, but not for IRF7 and Oasl2, whose expression was previously shown in mouse analysis. Moreover in Fig 2B HaCaTs were analyzed for IRF1, IRF9 and SOCS3 but therefor no data was shown in mice whereas mice were analysed for STAT2 and Ifit1 which was not the case in HaCaTs. Please also use the same please also use the same order of presentation in your graphics, this will make it easier to read and understand the graphics (in some cases this has already been done).

Our reply: We showed that almost all ISGs that we tested were regulated by FGF signaling in mouse and human keratinocytes. The basal expression of different ISGs and the efficiency of FGFs in their regulation was, however, slightly different for the mouse and human cells and the promoter regions of the mouse and human ISGs are also different. Therefore, we have not always used the same ISGs in our analysis. Nevertheless, we now extended our analysis and we show the results for different ISGs in a similar order.

2. Figure S2: Inhibition of Erk1/2, PI3K or Akt does not affect the FGF7-mediated suppression of ISG expression, while inhibition of proteasomal activity abrogates the FGF7 effect. Please insert a positive control for the inhibitor so that you can be sure that the inhibitor itself works.

Our reply: Positive controls for Erk1/2 or Akt inhibition had been shown in Fig. S2C and D, previous version (Western blot analysis of pErk1/2 and pAkt). Therefore, the activity of these inhibitors was confirmed. Similar experiments were performed for additional inhibitors that we tested for the revision. These results are now shown in Appendix Fig. S1. The activity of the proteasome inhibitors was confirmed by Western blot analysis of the NRF2 transcription factor, which is normally rapidly degraded via the ubiquitin/proteasome pathway. NRF2 protein levels strongly increased in the presence of all proteasome inhibitors. This result is now shown in Appendix Fig. S2B.

3. *Westernblots: it's not clear from the legend how high the n-number is. Means one experiment =n=1 or one experiment with n=?*

Our reply: We now explain this in more detail in the individual legends. In addition, we repeated some of these experiments and analyzed each condition in triplicate. The data were quantified (see Fig. 3C and 4B).

In order to increase the significance of the western blot, a densitometric evaluation of all westernblots should be carried out and their evaluation should be presented as a graph in addition to the existing exemplary image.

a. *Fig. 3C the differences in pSTAT1 activation between IFN α and IFN α +FGF7 are hardly to see, this may be improved by densitometric analysis (see above).*

b. *Fig. 4B the western blot for IRF3 and pIRF3 is not convincing . The manuscript also mentions STAT3, which is not shown in the figure.*

Our reply: The Western blot analyses of IFN- α - and Poly(I:C)-treated cells were repeated using triplicates for each condition and the data were quantified by densitometry. The new results are shown in Fig. 3C and 4B. STAT3 was not analyzed – this has been removed from the text.

c. *Fig. 6E the western blot is unfortunately not convincing because of the GapDH levels, which are much lower in the FGF7 stimulated cells than in the KCtrl or HSV-1 cells.*

Our reply: We agree that the GAPDH levels are lower in the protein lysates, which were also analyzed for Glyc-D. This even results in an underestimation of the effect of FGF7. In the other lysates there was no difference in the GAPDH levels between FGF7-treated and non-treated samples. To address this point we scanned the blots and we normalized each protein value to the GAPDH value of the same lysate. The results are now shown graphically in this figure.

4. *Fig 6G und follows: please explain the legend of the figure. Stain 976 of ZIKV? PF13/251013-18?*

Our reply: This has been done (see new legend to this figure)

5. *Fig7: Please control Fig 7. Some of the graphs are shifted.*

Our reply: We apologize for this problem, which seems to have happened during the file conversion. The figure displayed correctly in the final pdf document that was checked before submission.

6. Diskussion: Nectin-1 was shown to be involved in HSV-1 entry in keratinocytes (Charlotte L. Sayers and Gillian Elliott 2016). Is the expression of nectin-1 influenced by FGF treatment?

Our reply: As suggested by the reviewer we analyzed if nectin-1 expression is affected by FGF7 in keratinocytes. However, its expression was not up-regulated, but rather mildly down-regulated, strongly suggesting that promotion of viral infection by FGF7 is not mediated via nectin-1. These new data are now shown in Fig. 5J. We also cite the Sayers and Elliott paper as well as another paper (Petermann et al., 2015) that also showed an involvement of Nectin-1 in HSV-1 entry into keratinocytes.

7. Human keratinocytes: Mean n=6 that keratinocytes from 6 different donors were investigated or are they replicates? Please indicate it clearly in the legend.

Our reply: These were 6 replicates from two experiments with two donors. We now clarify this in the legend to Fig. 3D.

8. Statistical analysis: which test was used to test if the values come from a Gaussian distribution? Two-way ANOVA assumes that your replicates are sampled from Gaussian distributions, especially with small sample sizes this is important and should be taken into account.

Our reply: For comparison of more than two groups, we used Brown-Forsythe test to verify equal variance between groups, followed by one-way or two-way ANOVA and Tukey's multiple comparisons post-hoc tests.

Referee #2 (Remarks for Author):

Maddaluno and colleagues report the very interesting, important and original observation that signaling through FGFR1 and FGFR2 inhibits interferon stimulated gene (ISG) expression. They further show that this is functionally relevant in the context of viral infection. They propose that FGFR inhibitors could be used as broadly active antivirals.

The experiments are very well done, and the results support the conclusions. I have two major considerations:

1. The big missing part of this exciting story is the mechanism. How does FGF7 inhibit both tonic and induced ISG expression?

Our reply: We are aware that we have not fully explored the mechanism underlying this novel activity of FGFs, in particular since we focused on the consequences of the findings for viral infection/replication. Nevertheless, we already provided some insight into the mechanism: We showed that this effect is downstream of interferon receptors, that it occurs

at the transcriptional level, and that it requires proteasomal activity. In response to the comments of the reviewer we performed additional experiments (see below) and we now provide further mechanistic insight. However, we would like to point out that this is an entirely novel finding and it will clearly require extensive follow-up studies to further explore the molecular mechanisms.

The authors provide some insights with the proteasome inhibitor experiments (Figures 2G, 2H, S2G). They speculate that "The effect of FGF7 is likely to involve proteasomal degradation of a protein required for ISG expression, since it was abolished in the presence of different proteasome inhibitors." They further speculate that this mechanism could involve the well known ISG transcriptional regulators IRF9, IRF1, IRF7 and STAT1 and STAT2. ,There is indeed a slight reduction of the protein levels of this transcription factors (shown in figures 2C, 3C, and 6E) but in parallel there is a reduction of the corresponding mRNAs that could very well explain the reduced protein levels. The experiments with the proteasome inhibitor provide only circumstantial evidence. The authors should further explore this.

What is known about regulation of the proteasome pathway by FGF7?

Our reply: To our knowledge, there is as yet no published study describing an effect of FGF7 on the proteasome. A paper from 2006 (PMID 16720300) reported that FGF2 induces degradation of the phosphatase HD-PTP via the proteasome, but it was not determined if FGF2 affects proteasome activity.

In fact, our data do not necessarily imply that FGF7 activates the proteasome – rather, it may promote ubiquitination of one or more proteins, followed by their proteasomal degradation (see discussion). For example, FGFR activation was shown to activate the ubiquitin ligase c-Cbl, which is involved in receptor degradation. C-Cbl also targeted Sprouty 2 to the ubiquitin-dependent proteasome pathway in response to FGF stimulation of fibroblasts (PMID 12593796). We therefore tested if knock-down of c-Cbl affects the effect of FGF7 on ISGs, but this was not the case (our unpublished data). Therefore, it seems likely that other ubiquitin ligases are responsible, which need to be identified in an unbiased approach.

Although we favor an effect of FGF7 on protein ubiquitination, we had of course planned to test a potential direct effect of FGF7 on proteasome activity in keratinocytes as suggested by the reviewer. Our first attempts to address this issue failed due to a faulty kit, and the quality problem was confirmed by the company that sells the kit. An alternative kit was ordered, but due to the shut-down of our laboratories during the SARS-CoV-2 pandemic we were so far unable to perform these experiments. Therefore, we can unfortunately not provide these data at this stage.

Is the presumed FGF7 induced proteasome activation non-specific, or specifically/preferentially targeting ISG transcriptional regulators such as IRF7 and IRF1?

Our reply: As mentioned above, there is as yet no information on a potential activating effect of FGF7 on the proteasome. This effect may well be regulated at the level of ubiquitination. Nevertheless, it is of course very interesting to determine if the rapid suppression of gene expression by FGF7 and the rescue by proteasome inhibitors is specific for certain ISGs, for ISGs in general or if it also affects other genes (non-ISGs).

The inhibition of the FGF7-induced down-regulation by MG132 was not only observed for *IRF1* and *IRF7*, but also for *RSAD2*, *ISG15* and *STAT1* (other ISGs were not tested). We show

the *IRF1* and *IRF7* data in the manuscript, because we analyzed these genes in experiments with different proteasome inhibitors. We could certainly show the other genes as well upon request.

Our microarray data from isolated epidermis of ctrl and K5-R1/R2 mice demonstrate that other genes, which have not been described as ISGs, are upregulated in the FGFR-deficient mice, although the vast majority of the most highly regulated genes are ISGs. This is now mentioned on page 5, first paragraph. We have preliminary data showing that some of the other upregulated genes are directly (and negatively) regulated by FGFs. However, we do not yet know if they are also regulated via proteasomal degradation. The question regarding specificity for ISGs is indeed very interesting and important, but we feel that it goes beyond the scope of this initial publication. We will certainly test this in future studies.

What about mTORC1, a central regulator of proteasome assembly (and a potential target of FGF signaling downstream of Akt)?

Our reply: mTORC1 regulation by receptor tyrosine kinases usually occurs via PI3K/Akt signaling and neither inhibition of PI3K nor of Akt alone abolished the suppression of ISGs by FGF7.

Nevertheless and as suggested by the reviewer, we checked if inhibition of mTORC1 (by rapamycin) inhibits the FGF7-induced ISG suppression. We performed one experiment and the effect of FGF7 on two different ISGs was not affected. Unfortunately, we could not repeat the experiment because of the shut-down of our lab, but we have included the result of the first experiment as “additional information for reviewer 2” (first page).

The authors also provide some mechanistic insights by showing that blockade of the major FGFR signaling pathways, the PI3-kinase/Akt and Erk1/2 pathways, do not suppress the effect of FGF7 on ISG expression. They conclude that "other pathways or pathway combinations are involved." What about PLC γ or STATs? Is the FGFR kinase activity required for the observed effects on ISG expression?

Our reply: We also tested inhibitors for PLC γ , but they had no effect on the FGF7-mediated ISG suppression. These data are now shown in Fig. EV2D. Although we did not find a strong phosphorylation of STAT3 in keratinocytes upon FGF7 treatment (see for example “additional information for reviewer 2” (second page), we also tested a STAT3 inhibitor. However, it only marginally affected the FGF7-mediated ISG suppression. Since we could not repeat this experiment because of the lab shut-down, we provide this information as “additional information for reviewer 2” (second page). Most importantly, however, we include new data demonstrating that the combined inhibition of Erk1/2 and PI3K signaling inhibited the effect of FGF7. These new and repeated data are now shown in Fig. EV2C.

We already showed in Fig. 7A that FGFR kinase inhibitors suppress the effect of FGF7 on HSV-1 replication. We now also tested the effect of these inhibitors on FGF7-induced ISG expression in primary murine keratinocytes and in human HaCaT keratinocytes, and it was indeed inhibited. These new data are now shown in Fig. EV2A and B.

2. How generally valid is this observation? The authors focus their study on keratinocytes. What about other cells in the body? This seems an important question, since the authors

propose FGFR inhibition as an exciting new antiviral strategies, also for viruses that do not only replicate in keratinocytes.

Our reply: In response to this comment we also tested the effect of FGF7 on Caco-2 colon cancer cells, which express the high affinity receptor for FGF7 (FGFR2b) (PMID 19326389). Indeed, FGF7 also suppressed the expression of ISGs in these cells. These data are now shown in Fig. EV1B. In addition, we also mention in the Discussion that treatment of primary human lung epithelial cells reduced the expression of some ISGs (PMID 11459923). Therefore, the effect is not specific for keratinocytes, but occurs at least in other epithelial cells that express FGFR2b.

Specific points:

1. SOCS1 rather than SOCS3 is the canonical inhibitor of IFNAR signaling. Is SOCS1 regulated by FGFR2?

We checked the effect of FGF7 on *SOCS1* expression and its expression was also suppressed. These data are now shown in Fig. EV1A.

2. In many figures, qPCR values are shown as relative mRNA levels. Relative to what? Please indicate in each case in the figure legend or the figure.

Our reply: The reference genes had been specified in Materials and Methods. We now added this information to the Legends. We further specify in each legend that the mean value of the control samples was set to 1, and the relevant control is mentioned in each legend.

3. There is also another concern about showing these relative mRNA levels. For example in Figure 2D, Irf7 mRNA is relatively lower in IFNAR KO compared to wildtype. In Figure 2E, untreated wild type and IFNAR KO have the same mRNA level of Irf7 (around 1), and in both cases it goes down after FGF7 treatment. Showing all results in one figure would probably be preferable to avoid apparent inconsistencies between Fig 2D and 2E.

Our reply: We show these results separately, since the absolute expression levels in the knockout cells are of course much lower. Therefore, we would not be able to see the effect of FGF7 in a combined figure. However, we provide all raw data in Dataset EV2.

We would also like to point out that the mean expression level in non-treated WT cells in the left graph was set to 1 and in the other graphs the mean expression level in the non-treated IFNR knockout mice was set to 1. Therefore, the graphs show expression levels relative to non-treated control and not absolute expression levels. Thus, the results shown in Fig. 2D and 2E are not contradictory. We have further clarified this in the text and in the legend to this figure.

4. The figure legends are often un-sufficient. Length of treatments and concentration of reagents are often missing. Y axis description often missing. Authors should please go carefully through figures and legends.

Our reply: We had provided all concentrations in Materials and Methods, but we agree that the legends should also include this and additional information. Therefore, we extended the legends as requested. Y axis description has been included for all figures.

5. *The panels of Figure 7 are partially overlapping and seem to be dislocated.*

Our reply:

We apologize for this problem, which seems to have happened during the file conversion. The figure displayed correctly in the final pdf document that was checked before submission.

Referee #3 (Comments on Novelty/Model System for Author):

frankly i am not too confident about the therapeutic applications of these findings

Our reply: We show that FGF7 regulates replication of different viruses. We now further show that this effect is not restricted to keratinocytes. Therefore, our findings may well have therapeutic implications (as also acknowledged by the other reviewers). Nevertheless, we now discuss this more carefully (see last point of the reviewer).

Referee #3 (Remarks for Author):

This MS reports the finding that Mice with a KO of FGF receptors (FGFR) 1 and 2 in their skin have high levels of expression of IFN stimulated genes (ISG). Further investigation of this initial observation leads to the conclusion that FGF signaling inhibits the expression of ISGs in mouse and human keratinocytes and, as such, promotes the replication of HSV1 and other viruses in the same cells. This notion is supported by a variety of experiments using FGF7 treatment of Keratinocytes, by showing that ISG expression is stimulated by FGFR inhibitors, and that FGF signaling competes with IFN or Poly:IC treatment in regulating ISG expression. Finally it is shown that the replication of HSV1, LMC and zika viruses is promoted by FGF signaling by dampening the IFN response and that inhibition of FGF signaling adversely affects HSV1 replication.

The results presented here are quite convincing and the major conclusion of the MS, i.e. that FGF signaling antagonizes IFN signals is amply documented.

The major weakness of this report is however that it does not identify or even investigate the mechanisms responsible for the phenomenon described.

Our reply: We are aware that we have not fully explored the mechanism of this novel finding, in particular since we focused on the consequences of the FGF7-mediated suppression of IFN and ISGs for viral replication. Nevertheless, we already provide some insight into the mechanism: We show that this effect is downstream of the interferon receptors, that it occurs at the transcriptional level, and that it requires proteasomal activity. In response to the comments of reviewers 2 and 3 we performed additional experiments and we now provide further mechanistic insight (see below). However, we would like to point out that this is an entirely novel finding and it will clearly require extensive follow-up studies to further explore the molecular mechanisms.

To address the concern of the reviewer, we performed additional experiments and we now show that the effect of FGF7 is dependent on FGFR kinase activity. Furthermore, we show

that combined inhibition of Erk1/2 and PI3K abolishes the effect of FGF7 on ISG expression, while inhibition of other pathways had no effect. These new data are now shown in Fig. EV2.

Furthermore I noticed that, in most experiments testing the expression of the ISGs in response to FGF treatment (figs. 1,2,3,4, 6), the effect on the RNA levels of the genes are much stronger than the effect on protein levels, that in some cases show no difference.

Our reply: We repeated the Western blot experiments using triplicates from each condition and we quantified the data. These data are now shown in Fig. 3C and 4D. We agree that the effect on the RNA is often stronger than on the protein level, but this was also dependent on the time point. Nevertheless, we clearly observed significant effects of FGF7 on IRF1 and IRF9 in the basal state and a strong suppression of the poly(I:C)-induced STAT1 and STAT2 phosphorylation by FGF7.

Unfortunately the genes whose expression is determined at the RNA level and those whose protein expression is shown are often not the same, making difficult to ascertain how general is this phenomenon. This discrepancy has to be discussed or explained if this MS has to be accepted for publication.

Our reply: The ISG products that we show in the Western blots are also regulated at the transcriptional level by FGF7. Unfortunately, we could not analyze the protein levels of all ISGs that are regulated at the RNA level because of the insufficient quality of the antibodies. Nevertheless, we believe that it is sufficient to show a selection of ISG products at the protein level.

Two minor points: 1) it would be better to measure viral titers, in Infectious units, rather than DNA levels.

Our reply: We show viral titers in Fig. 7C and they strongly correlated with viral DNA and protein levels.

2)The Ms is extremely long and could be trimmed considerably. The discussion is rambling, never discusses mechanisms and some of the therapeutic implications could be toned down.

Our reply: As requested by the reviewer, we shortened the manuscript, included a discussion about mechanisms and toned down the potential therapeutic implications.
toned down the potential therapeutic implications.

17th Jun 2020

Dear Sabine,

Thank you for the submission of your revised manuscript to EMBO Molecular Medicine. We have now received the enclosed reports from the referees that were asked to re-assess it. As you will see the reviewers are now globally supportive and I am pleased to inform you that we will be able to accept your manuscript pending the following final amendments:

Please address the minor comments of referee 1 and address the statistical questions to the best of your ability. I'm not sure whether you will be able to add more primary cells analysis from a different donor, if you don't still make sure to indicate the number of donors in the legend and discuss the limitation of data as obtained from only 2 donors.

Please provide a point-by-point letter INCLUDING my comments as well as the reviewer's reports and your detailed responses to their comments (as Word file).

Please submit your revised manuscript within two weeks.

I look forward to reading a new revised version of your manuscript as soon as possible.

Yours sincerely,

Celine Carret

Celine Carret, PhD
Senior Editor
EMBO Molecular Medicine

*** Instructions to submit your revised manuscript ***

To submit your manuscript , please follow this link:

Link Not Available

- 1) a .docx formatted version of the manuscript text (including Figure legends and tables)
- 2) Separate figure files*
- 3) supplemental information as Expanded View and/or Appendix. Please carefully check the authors guidelines for formatting Expanded view and Appendix figures and tables at <https://www.embopress.org/page/journal/17574684/authorguide#expandedview>
- 4) a letter INCLUDING the reviewer's reports and your detailed responses to their comments (as Word file).

***** Reviewer's comments *****

Referee #1 (Comments on Novelty/Model System for Author):

the authors have confirmed the results in the human keratinocyte cell line HaCaT and have also conducted individual experiments in primary keratinocytes. In addition, functional assays were performed to confirm the biological relevance of their data.

Referee #1 (Remarks for Author):

Most of the concerns were well taken up by the authors. However, there are still some minor points that the authors should address:

1. Of course, there are cell type and species-specific differences in the regulation of ISG by FGF. But these differences are also of interest, especially for the assessment of a possible potential of FGF receptor inhibitors as therapeutic targets for viral infections. Even if no common panel of ISGs for mouse and human was measured/is shown, these differences should at least be mentioned and discussed in the manuscript.

2. In Figure 4B the graph showing the quantification of western blot analysis is missing.

3. Using primary human cells, the number of donors should be increased to get a robust power of your findings. The number of two different donors is too low. Furthermore, you should at least perform 3 independent experiments. Making statistics from only two donors even they were measured as triplicates has no power. The number of different donors (primary human keratinocytes) should be added to each figure legend.

Referee #2 (Remarks for Author):

The authors answered my questions and concerns in their point-to-point reply and improved the manuscript.

The authors performed the requested editorial changes.

Reviewer's comments**Referee #1 (Comments on Novelty/Model System for Author):**

The authors have confirmed the results in the human keratinocyte cell line HaCaT and have also conducted individual experiments in primary keratinocytes. In addition, functional assays were performed to confirm the biological relevance of their data.

Referee #1 (Remarks for Author):

Most of the concerns were well taken up by the authors. However, there are still some minor points that the authors should address:

1. Of course, there are cell type and species-specific differences in the regulation of ISG by FGF. But these differences are also of interest, especially for the assessment of a possible potential of FGF receptor inhibitors as therapeutic targets for viral infections. Even if no common panel of ISGs for mouse and human was measured/is shown, these differences should at least be mentioned and discussed in the manuscript.

Our reply: This has been done (see page 7, first paragraph). We would also like to point out that we verified the effect of FGF7 on almost all ISGs that we showed to be regulated in mouse keratinocytes also with human HaCaT keratinocytes. We just did not analyze all ISGs in all experiments, because the regulation of the different ISGs was so well reproducible. The only exception was *Oasl2*, because the human orthologue *OASL* is quite different with regard to sequence and function (Eskildsen et al., 2003). We also mention on page 7, last paragraph and page 8, first paragraph, that *RSAD2* was not expressed in Caco-2 cells and that *IRF7* expression was not regulated by FGF7. This shows the cell-type specific differences in ISG expression.

2. In Figure 4B the graph showing the quantification of western blot analysis is missing.

Our reply: The graph was/is shown in Figure EV3 as mentioned in the legend to Fig. 4B.

3. Using primary human cells, the number of donors should be increased to get a robust power of your findings. The number of two different donors is too low. Furthermore, you should at least perform 3 independent experiments. Making statistics from only two donors even they were measured as triplicates has no power.

We agree that only two donors were used in the experiments shown in Fig. 3D and 5E, but the data points obtained with both donors clustered very well in both cases. The two donors were analyzed in independent experiments and therefore, the results were well reproducible. Furthermore, the human primary keratinocytes used in the experiments shown in Fig. 3D and 5E were from different donors. We now mention the limitation in the Results (page 10, first paragraph) and we also mention the number of donors in the legends (this was already done in the previous version for the results shown in Fig. 3D). We would like to point out that the data were also reproduced with primary mouse keratinocytes and with HaCaT keratinocytes and they were all consistent. Finally, we used primary

keratinocytes from a third donor and showed that poly(I:C)-induced IFN expression is suppressed by FGF7 (so far performed with cells from one donor, and therefore not included in the manuscript). We show this result below for the information of the reviewer.

Figure for reviewers removed

It was unfortunately not possible to obtain cells from a third donor for a third repetition within the 2-week time frame that was given for the revision, but we believe that the results described above clearly show the reproducibility of the FGF7 effect with different types of keratinocytes and with cells from different donors.

The number of different donors (primary human keratinocytes) should be added to each figure legend.

Our reply: The number of donors had already been mentioned in the legend to Fig. 3D. We now included this information in the legend to Fig. 5E.

Referee #2 (Remarks for Author):

*The authors answered my questions and concerns in their point-to-point reply and improved the manuscript.
ms and turned down the potential therapeutic implications.*

Our reply: We thank the reviewer for his/her positive comments.

30th Jun 2020

Dear Sabine,

Thank you for amending the article. We are very pleased to inform you that your manuscript is accepted for publication and is now being sent to our publisher to be included in the next available issue of EMBO Molecular Medicine.

Please read below for additional IMPORTANT information regarding your article, its publication and the production process.

Congratulations on your interesting work and all the best,

Celine

Celine Carret, PhD
Senior Editor
EMBO Molecular Medicine

Follow us on Twitter @EmboMolMed
Sign up for eTOCs at embopress.org/alertsfeeds

Corresponding Author Name: Luigi Maddaluno and Sabine Werner

Manuscript Number: EMM-2019-11793